


# Towards standardized processing of eddy covariance flux measurements of carbonyl sulfide

Kukka-Maaria Kohonen[1], Pasi Kolari[1], Linda M.J. Kooijmans[2], Huilin Chen[3], Ulli Seibt[4], Wu Sun[4,5], and Ivan Mammarella[1]

[1]Institute for Atmospheric and Earth System Research (INAR)/ Physics, Faculty of Science, University of Helsinki, Helsinki, Finland
[2]Meteorology and Air Quality group, Wageningen University and Research, Wageningen, the Netherlands.
[3]Centre for Isotope Research (CIO), University of Groningen, Groningen, the Netherlands
[4]Department of Atmospheric and Oceanic Sciences, University of California, Los Angeles, California, USA
[5]Department of Global Ecology, Carnegie Institution for Science, Stanford, California, USA

**Correspondence:** Kukka-Maaria Kohonen (kukka-maaria.kohonen@helsinki.fi)

**Abstract.** Carbonyl sulfide (COS) flux measurements with the eddy covariance (EC) technique are growing in popularity with the recent development in using COS to estimate gross photosynthesis at the ecosystem scale. Flux data intercomparison would benefit from standardized protocols for COS flux data processing. In this study, we analyze how various data processing steps affect the final flux and provide a method for gap-filling COS fluxes. Different methods for determining the lag time between
COS mixing ratio and the vertical wind velocity ($w$) resulted in a maximum of 12 % difference in the cumulative COS flux. Due to limited COS measurement precision, small COS fluxes (below approximately 3 pmol m$^{-2}$ s$^{-1}$) could not be detected when the lag time was determined from maximizing the covariance between COS and $w$. We recommend using a combination of COS and carbon dioxide (CO$_2$) lag times in determining the COS flux, depending on the flux magnitude compared to the detection limit of each averaging period. Different high frequency spectral corrections had a maximum effect of 10 % on COS
flux calculations and different detrending methods only 1.2 %. Relative total uncertainty was more than five times higher for low COS fluxes (lower than $\pm 3$ pmol m$^{-2}$s$^{-1}$) than for low CO$_2$ fluxes (lower than $\pm 1.5$ $\mu$mol m$^{-2}$s$^{-1}$), indicating a low signal-to-noise ratio of COS fluxes. Due to similarities in ecosystem COS and CO$_2$ exchange, and the low signal-to-noise ratio of COS fluxes that is similar to methane, we recommend a combination of CO$_2$ and methane flux processing protocols for COS EC fluxes.

## 1   Introduction

Carbonyl sulfide (COS) is the most abundant sulfur compound in the atmosphere, with tropospheric mixing ratios around 500 ppt. During the last decade, studies on COS have grown in number, mainly due to the use of COS exchange as a tracer for photosynthetic carbon uptake (also known as gross primary production, GPP) (Sandoval-Soto et al., 2005; Blonquist et al.,





2011; Asaf et al., 2013). COS shares the same diffusional pathway in the leaves as carbon dioxide ($CO_2$), and COS is destroyed completely by hydrolysis and not being emitted. This uni-directional flux makes it a promising proxy for GPP, particularly in light of recent studies that account for radiation-dependency of COS to $CO_2$ uptake rates (Yang et al., 2018; Kooijmans et al., 2019).

Eddy covariance (EC) measurements are the backbone of gas flux measurements at the ecosystem scale (Aubinet et al.,
2012). Protocols for instrument setup, monitoring and data processing have been established especially for $CO_2$ (Rebmann et al., 2018; Sabbatini et al., 2018) but recently also for other greenhouse gases, such as methane ($CH_4$) and nitrous oxide ($N_2O$) (Nemitz et al., 2018), due to the growing network of the Integrated Carbon Observation System (ICOS) flux stations (Franz et al., 2018).

To meet the assumptions for EC measurements, the experimental setup needs to be carefully designed. The site should be
homogeneous, so that the footprint area solely consists of the ecosystem type of interest, and the measurement height should be chosen such that it is well above any obstacles (such as canopy top). Moreover, measurements should be made in the well-mixed surface layer but still low enough that the stable atmosphere would not extend the footprint away from the desired location (Aubinet et al., 2012). Both the gas analyzer and sonic anemometer need to operate at a high frequency, usually 10 or 20 Hz, to capture the fast turbulent fluctuations.

Gas analyzers used for EC measurements can be divided into open-path analyzers, where gas analysis happens in the open air, and closed-path analyzers, where the gas sample is pumped into an enclosed sample cell through a sampling tube. Closed-path analyzers need filters in the sampling line to prevent clogging of sample tubing and contamination of instrument components. One coarser filter is recommended to be placed close to the inlet while a finer filter should be placed close to the analyzer (Nemitz et al., 2018). The instrumentation needs regular checks to maintain data quality and to prevent long measurement
gaps. Other considerations include, e.g., regular cleaning or changing of the sampling lines to prevent adsorption or desorption of water vapor on tubing walls (Mammarella et al., 2009).

Standardized protocols have been developed for $CO_2$ EC data processing, which follows the steps detailed in Sabbatini et al. (2018): despiking and filtering raw data to eliminate spikes and erroneous data, coordinate rotation of the wind components according to the prevailing wind direction, determining the time lag between the sonic anemometer and the gas analyzer
signals, trend removal (to separate the turbulent fluctuations from the mean trend), calculating covariances and correcting for flux losses at low and high frequencies. Flux post-processing then includes quality filtering and quality flagging according to atmospheric turbulence characteristics and stationarity.

During nighttime, decoupling layers may form between the measurement height and the surface in a stable atmosphere. In conditions of low atmospheric turbulence, the EC flux is usually underestimated and thus needs to be either removed or
corrected. The nighttime problem is well studied in the EC community (Aubinet et al., 2003, 2005, 2010) but not solved yet. In addition to decoupling layers, advective transport may become important in stable conditions. Vertical advection has been used to explain the nighttime flux observations (Lee, 1998; Mammarella et al., 2007; Rannik et al., 2009), but with the lack of horizontal advection measurements is not recommended as a correction anymore (Aubinet et al., 2003). Storage change flux correction and friction velocity ($u_*$) filtering are the most commonly used methods for correcting and filtering out low-



turbulence occasions. Stable stratification may also enlarge the flux footprint so that it extends across other surface cover types. The final step of the post-processing is then filling the gaps in the data using gap-filling functions based on environmental data.

Studies on ecosystem COS flux measurements with the eddy covariance (EC) technique are still limited (Asaf et al., 2013; Billesbach et al., 2014; Commane et al., 2015; Wehr et al., 2017; Yang et al., 2018; Kooijmans et al., 2019; Spielmann et al., 2019) and there is no standardized flux processing protocol for COS EC fluxes. Currently there are two commercial models

of closed-path spectroscopic analyzers for measuring COS at 10 Hz from Aerodyne Research (Billerica, MA, USA) and Los Gatos Research (San Jose, CA, USA), respectively. COS EC flux measurements and data processing have similarities with other trace gases (e.g., $CH_4$ and $N_2O$) that have low signal-to-noise ratios, especially regarding lag time determination and frequency response corrections. Accurate lag time determination is essential for correctly synchronizing the wind and gas concentration measurements, a basis of EC measurements, and an incorrect lag time would bias the flux estimates. Frequency

response corrections, on the other hand, are needed for correcting signal losses both at high and low frequencies and are always increasing the fluxes (Aubinet et al., 2012). However, unlike for $CH_4$ or $N_2O$, there are no sudden bursts or sinks expected for COS, and in that sense some of the processing steps are more like those for $CO_2$ (e.g., despiking, storage change correction and $u_*$ filtering). Gerdel et al. (2017) describes the issues of different detrending methods and high-frequency spectral correction. However, there has not been any study on the lag time determination and the nighttime measurement issues. This weakens our

ability to assess uncertainties in COS flux measurements.

In this study, we provide suggestions for a robust EC flux measurement setup for COS studies, give recommendations for processing COS EC flux data and discuss the most important sources of random and systematic errors. In addition, we introduce a method for gap-filling COS fluxes for the first time.

## 2 Materials and Methods

In this study we utilized COS and $CO_2$ EC flux datasets collected in Hyytiälä ICOS station in Finland from 26 June to 2 November 2015. The site has a long history of flux and concentration observations (Hari and Kulmala, 2005) and a COS analyzer was introduced to the site in March 2013. In this section, we go through all processing steps in EC flux processing. In the next section, the different processing schemes are compared to a "standard scheme", which consists of linear detrending, 2D wind rotation, using $CO_2$ lag time for COS and experimental spectral correction. The dataset used in this study was partly

published by Kooijmans et al. (2017), where only the nighttime data is utilized with the standard processing scheme.

### 2.1 Site description

Measurements were made in a boreal Scots pine (*Pinus sylvestris*) stand in the Station for Measuring Forest Ecosystem-Atmosphere Relations (SMEAR II) in Hyytiälä, Finland (61°51′ N, 24°17′ E, 181 m above sea level). The Scots pine stand was established in 1962 and reaches at least 200 m to all directions and about 1 km to the north (Hari and Kulmala, 2005). The

site is characterized by modest height variation and an oblong lake is situated about 700 m to the southwest of the forest station (Rannik, 1998; Vesala et al., 2005). Canopy height was 17 m and all-sided leaf area index (LAI) approximately 8 $m^2m^{-2}$ in





2015. EC measurements were done at 23 m height. The site became a certified ICOS class 1 atmospheric station in 2017 and class 1 ecosystem station in 2018. Sunrise time varied from 2:37 am in June to 7:55 am in November, while sunset was at 10:14 pm in the beginning and 4:17 pm in the end of the measurement period (all times presented in UTC+2, local winter time).

## 2.2 EC measurement setup

The EC setup consisted of an ultrasonic anemometer (Solent Research HS1199, Gill Instruments Ltd., England, UK) for measuring wind speed in three dimensions, an Aerodyne quantum cascade laser spectrometer (QCLS) (Aerodyne Research Inc., Billerica, USA) for measuring COS, $CO_2$, carbon monoxide (CO) and water vapor ($H_2O$) mole fractions and an LI-6262 infrared gas analyzer (Licor, Nebraska, USA) for measuring $H_2O$ and $CO_2$ mole fractions. All measurements were done at 10 Hz frequency and with a flow rate of approximately 10 liter per minute (LPM) for the QCLS. The PTFE sampling tube was 32 m long and had an inner diameter of 8 mm. Two PTFE filters were used upstream of the instrument inlet to prevent any contaminants entering the analyzer sample cell: one coarse filter (0.45 $\mu$m, Whatman), followed by a finer filter (0.2 $\mu$m, Pall corporation), at approximately 50 cm distance to the analyzer inlet. The Aerodyne QCLS uses an electronic pressure control system to control the pressure fluctuations in the sampling cell. The QCLS was running at 20 Torr sampling cell pressure. An Edwards XDS35i scroll pump (Edwards, England, UK) was used to pump air through the sampling cell.

Background measurements of high-purity nitrogen ($N_2$) were done every 30 min for 26 s to remove background spectral structures in the QCLS (Kooijmans et al., 2016). In addition, a calibration cylinder was sampled each night at 00:00:45 for 15 s. The calibration cylinder consisted of COS at 429.6 $\pm$ 5.6 ppt, $CO_2$ at 408.37 $\pm$ 0.05 ppm and CO at 144.6 $\pm$ 0.2 ppb. The cylinder was calibrated against the NOAA-2004 COS scale, WMO-X2007 $CO_2$ scale and WMO-X2004 CO scale using cylinders that were calibrated at the Center for Isotope Research of the University of Groningen in the Netherlands (Kooijmans et al., 2016). The standard deviation was 19 ppt for COS mixing ratios and 1.3 ppm for $CO_2$ at 10 Hz measurement frequency, as calculated from the cylinder measurements.

It has previously been shown that water vapor in the sample air can affect the measurements of COS through spectral interference of the COS and $H_2O$ absorption lines (Kooijmans et al., 2016). This spectral interference was corrected for by fitting the COS spectral line separately from the $H_2O$ spectral line.

The computer embedded in the Aerodyne QCLS and the computer that controlled the sonic anemometer were synchronized once a day with a separate server computer. Data were logged in separate files with a home-made software (COSlog).

## 2.3 Profile measurements

Atmospheric concentration profiles were measured with another Aerodyne QCLS at a sampling frequency of 1 Hz. Air was sampled at 5 heights: 125 m, 23 m, 14 m, 4 m and 0.5 m. A multi-position Valco valve (VICI, Valco Instruments Co. Inc.) was used to switch between the different profile heights and calibration cylinders. Each measurement height was sampled for 3 min each hour. One calibration cylinder was measured twice for 3 min each hour to correct for instrument drift, and two other calibration cylinders were measured once for 3 min each hour to assess the long-term stability of the measurements. A background spectrum was measured once every six hours using high-purity nitrogen (N 7.0) (for more details, see Kooijmans





et al. (2016)). The overall uncertainty of this analyzer was determined to be 7.5 ppt for COS and 0.23 ppm for $CO_2$ at 1 Hz frequency (Kooijmans et al., 2016). The measurements are described in more detail in Kooijmans et al. (2017).

## 2.4 Eddy covariance fluxes

In this section, we go through the processing steps of EC flux calculation from raw data handling to final flux gap-filling and uncertainties. Fig. 1 provides a graphical outline of all processing steps. The different processing options presented here are
applied and discussed in Sec. 3.

### 2.4.1 Pre-processing

For flux calculation, the sonic anemometer and gas analyzers need to be synchronized. The following procedure was done to combine two data files of 30 min length (of which one includes sonic anemometer and LI-6262 data and the other includes Aerodyne QCLS data): 1) QCLS data were first forced to have the same time lag as LI-6262 $CO_2$, 2) then the cross-covariance
of the two $CO_2$ signals (QCLS and LI-6262) was calculated 3) finally the QCLS data were shifted so that the cross-covariance of the $CO_2$ signals was maximized. The time shift was maximum 10 seconds, most often varying between 0 s to 2 s during one day.

Raw data were then despiked so that the difference between subsequent data points was maximum 5 ppm for $CO_2$, 1 mmol $mol^{-1}$ for $H_2O$, 12 ppb for CO and 200 ppt for COS. After despiking, the missing values were gap-filled by linear interpolation.
We used the 2D coordinate rotation to rotate the coordinate frame so that the turbulent flux divergence is as close as possible to the total flux divergence. First, the average $u$-component was forced to be along the prevailing wind direction. The second rotation was performed to force the mean vertical wind speed ($\bar{w}$) to be zero (Kaimal and Finnigan, 1994). In this way, the $x$-axis is parallel and $z$-axis perpendicular to the mean flow. In addition, for determining the vertical advection and flux uncertainties (described in detail later in the text), we used the planar fit method for coordinate rotation. In this method, $\bar{w}$ is
assumed to be zero only on longer time scales (weeks or even longer). A mean streamline plane is fitted to a long set of wind measurements. Then the $z$-axis is fixed as perpendicular to the plane and $\bar{v}$ wind component to be zero (Wilczak et al., 2001). While 2D coordinate rotation is the most commonly used rotation method, the planar fit method (even though more demanding by computation) brings benefits especially in complex terrain (Lee and Finnigan, 2004).

To separate the mixing ratio time series into mean and fluctuating parts, we tested three different detrending options: 1) 30
min block averaging (BA), 2) linear detrending (LD) and 3) recursive filtering (RF) with a time constant of 30 s. BA is the most commonly used method for averaging the data with the benefit of dampening the turbulent signal the least. On the other hand, it may lead to an overestimation of the fluctuating part (and thus overestimating the flux), especially due to instrumental drift or large scale changes in atmospheric conditions (Aubinet et al., 2012). The LD method fits a linear regression to the averaging period and thus gets rid of instrumental drift and to some extent of weather changes, but may lead to underestimation of the
flux if the linear trend was related to actual fluctuations in the atmosphere. The third method, RF, uses a time window (here 30 s) for a moving average over the whole averaging period. RF brings the biggest correction and thus lowest flux estimate compared to other methods, but effectively removes biased low-frequency contributions to the flux. In the ICOS protocol for



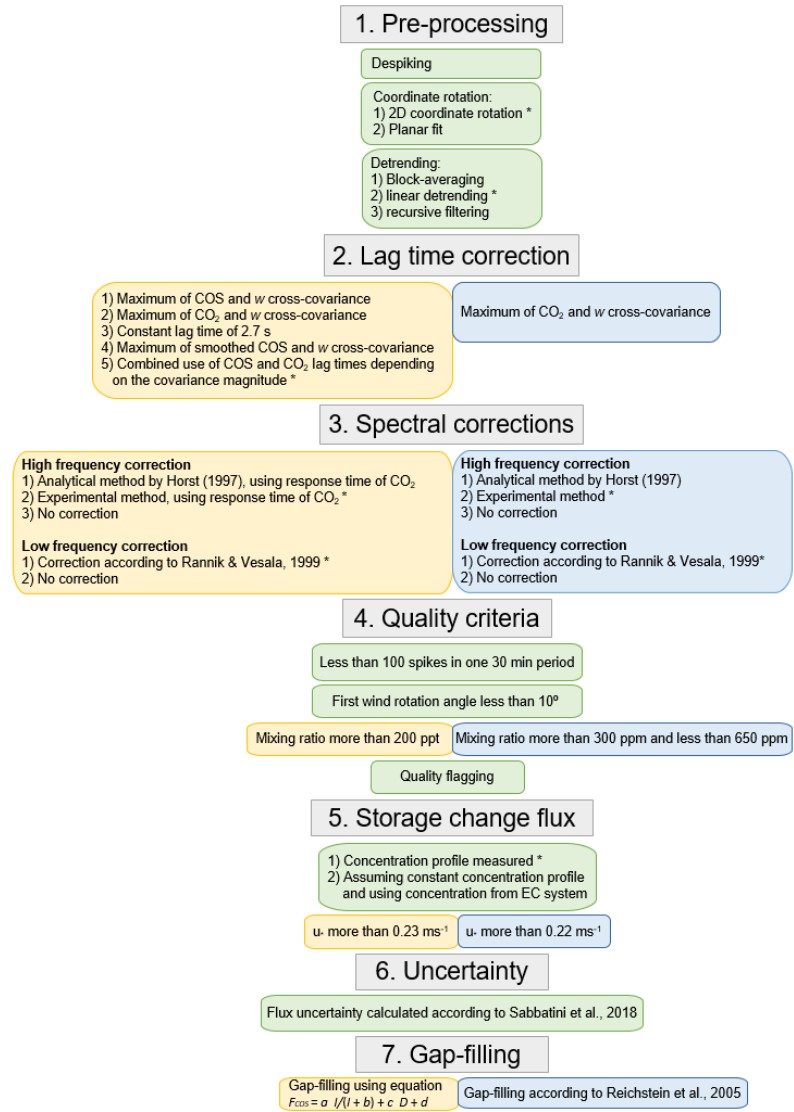

**Figure 1.** Different EC processing steps summarized. Yellow boxes refer to steps only used for COS data processing, blue to steps used only to $CO_2$ data and green boxes to steps that are relevant for both gases. Recommended options are marked with a star. Standard options that are used as a reference for COS in this study are: 2D coordinate rotation, linear detrending, combined use of COS and $CO_2$ lag times, experimental high frequency correction, low frequency correction according to Rannik and Vesala (1999) and storage change flux from measured concentration profile. Advection correction (that was ignored in this study) would follow storage change flux calculation.



small fluxes, it is recommended to use BA for estimating the background signal (Nemitz et al., 2018). The effect of different detrending methods will be shown in Sect. 3.

### 2.4.2 Lag time determination

The lag time was determined using the following five different methods:

1) From the maximum cross-covariance of the COS mixing ratio and $w$ $(\overline{w'\chi'_{COS}})$ within lag window 1.5–3.8 s (referred hereafter as COS lag). Lag window limits were determined based on the nominal lag time of 2.6 s calculated from the flow rate and tube dimensions. More flexibility was given to the longer end of the lag window as lag times have been found to be longer than the nominal lag time (Massman, 2000; Gerdel et al., 2017).

2) From the maximum cross-covariance of the $CO_2$ mixing ratio and $w$ $(\overline{w'\chi'_{CO_2}})$ within lag window 1.5–3.8 s (referred hereafter as $CO_2$ lag).

3) Using a constant lag time of 2.7 s, which is the most common lag for $CO_2$ with our setup (referred hereafter as Const lag).

4) From the maximum of smoothed $\overline{w'\chi'_{COS}}$ when the cross-covariance was smoothed with a 1.1 s running mean (referred hereafter as RM lag). The averaging window was chosen so that it provided a more distinguishable covariance maximum while still preventing a shift in the timing of the maximum.

5) A combination of COS and $CO_2$ lag times. First, the random flux error due to instrument noise was calculated according to Mauder et al. (2013):

$$RE = \sqrt{\frac{(\sigma_c^{noise})^2 \sigma_w^2}{N}} \tag{1}$$

where instrumental noise $\sigma_c^{noise}$ was estimated from the method proposed by Lenschow, D. and Wulfmeyer, V. (2000), $\sigma_w$ is the standard deviation of the vertical wind speed and $N$ the number of data points in the averaging period. The random error was then compared to the raw maximum covariance. If the maximum covariance was higher than three times the random flux error, then the COS lag method was used for lag time determination. If the covariance was dominated by noise (the random error being smaller than three times the random error) then the $CO_2$ lag method was selected, as proposed in (Nemitz et al., 2018) (referred hereafter as the DetLim lag).

### 2.4.3 Frequency response correction

Some of the turbulence signal is lost both at high and low frequencies due to losses in sampling lines, inadequate frequency response of the instrument and inadequate averaging times, among others (Aubinet et al., 2012). In this section, we describe both high and low frequency loss corrections in detail.

**High-frequency correction**

Especially closed-path systems cause the fluctuations to dampen at high frequencies due to long sampling lines. Other reasons for high-frequency losses include sensor separation and inadequate frequency response of the sensor. In turn, high-frequency





losses cause the normalized co-spectrum of the gas with $w$ to be lower than expected at high frequencies, resulting in lower

flux. In this study, we tested the effect of high-frequency spectral correction by applying either an analytical correction for high-frequency losses (Moore, 1986; Horst, 1997) or an experimental correction. The analytical correction was based on scalar co-spectra defined in Horst (1997) and the experimental approach was based on the assumption that temperature co-spectrum is measured without significant error and the normalized scalar co-spectra were compared to the normalized temperature co-spectrum (Aubinet et al., 2000; Mammarella et al., 2009).

Analytical high frequency attenuation for the scalar fluxes were determined by a transfer function by Horst (1997):

$$H_{ws}(n) = [1 + (2\pi\tau_s f_m)^\alpha]^{-1} \tag{2}$$

where $\alpha$ = 7/8 for neutral and unstable stratification and $\alpha$ = 1 for stable stratification in the surface layer. $\tau_s$ is the sensor specific time constant (0.68 s in our case for the Aerodyne QCLS and 0.032 s for LI-6262) and $f_m$ is the frequency of the logarithmic co-spectrum peak estimated from $f = \dfrac{n_m \bar{u}}{z - d}$ , where $n_m$ is the normalized frequency of the co-spectral peak, $\bar{u}$

wind speed, $z$ the measurement height and $d$ the displacement height. The normalized frequency of the co-spectral peak $n_m$ is dependent on stability $\zeta = \dfrac{z - d}{L}$ (Horst, 1997):

$$n_m = 0.085, \text{ for } \zeta \leq 0 \tag{3}$$

$$n_m = 2.0 - 1.915/(1 + 0.5z/L), \text{ for } \zeta > 0 \tag{4}$$

where $z$ is the measurement height and $L$ is the Monin-Obukhov length.

Empirical estimation of co-spectral transfer functions and flux attenuation was done according to Mammarella et al. (2009) using a site-specific co-spectral model, as recommended in De Ligne et al. (2010). The transfer function, which describes the attenuation, was defined as the normalized ratio of the cospectral densities

$$T_{ws} = \frac{N_\theta C_{ws}(f)}{N_s C_{w\theta}(f)} \tag{5}$$

where $N_\theta$ and $N_s$ are normalization factors and $C_{ws}$ and $C_{w\theta}$ the scalar and temperature cospectra, respectively. A sigmoidal

function $\dfrac{1}{1 + (2\pi f\tau_s)^2}$ was fitted to the transfer function and $\tau_s$ determined as the parameter of the least squares regression.

**Low-frequency correction**

Detrending the turbulent time series, especially with LD or RF methods, may also remove part of the real low frequency variations in the data (Lenschow et al., 1994; Kristensen, 1998), which should be corrected for in order to avoid flux underes-

timation. Low frequency correction in this study for different detrending methods was done according to Rannik and Vesala (1999). One run was performed without both low frequency and high frequency response corrections.

### 2.4.4   Flux quality criteria

The calculated fluxes pass the quality criteria when: the wind rotation angle ($\theta$) is below 10 °, the number of spikes in one half hour is less than 100, the COS mixing ratio is higher than 200 ppt, the $CO_2$ mixing ratio ranges between 300 ppm and 650 ppm

and the $H_2O$ mixing ratio is higher than 1 ppb.





For quality flagging, the standard criteria by Sabbatini et al. (2018) were used both for COS and $CO_2$: flag 0 was given if flux stationarity was less than 0.3 (meaning that covariances calculated over 5 min intervals deviate less than 30 % from the 30 min covariance), kurtosis was between 1 and 8 and skewness was within a range from -2 to 2. Flag 1 was given if flux stationarity was from 0.3 to 1 and kurtosis and skewness were within the ranges given earlier. Flag 2 was given if these criteria were not
met.

In addition to these filtering and flagging criteria, we added friction velocity ($u_*$) filtering to screen out low turbulence. However, it has to be kept in mind that the assumptions under which the $u_*$ filtering are applied (that fluxes do not go down under low turbulence conditions) may not be justified for COS (Kooijmans et al., 2017), as will be further discussed in Sect. 3.6. The appropriate $u_*$ threshold was derived from 99 % threshold criterion (Papale et al., 2006; Reichstein et al., 2005). The
lowest acceptable $u_*$ value was determined from both COS and $CO_2$ nighttime fluxes.

### 2.4.5    Storage change flux calculation

Storage fluxes were calculated from concentration profile measurements and from EC system concentration measurements by assuming a constant concentration profile throughout the canopy. Storage change fluxes from concentration profile measurements were calculated using the formula

$$F_{stor} = \frac{p}{RT_a} \int_0^h \frac{\partial \chi_c(z,t)}{\partial t} dz,$$                                                                                      (6)

where $p$ is the atmospheric pressure, $T_a$ air temperature, $R$ the universal gas constant, $h$ the EC measurement height (23 m) and $\chi_c(z)$ the gas mixing ratio at each measurement height. The integral was determined from hourly measured profile concentrations at 0.5 m, 4 m, 14 m, and 23 m by integrating an exponential fit through the data (Kooijmans et al., 2017). When the profile measurement was not available, storage was calculated from COS (or $CO_2$) concentration measured by the
EC setup.

Another storage flux calculation was done assuming a constant profile from the EC measurement height (23 m) to the ground level. A running average over a 5 hour window was applied to the COS concentration data to reduce the random noise of COS concentration measurements.

The storage fluxes were used to correct the EC fluxes for storage change of COS and $CO_2$ below the flux measurement
height.

### 2.4.6    Flux uncertainty

The flux uncertainty was calculated according to ICOS recommendations presented by Sabbatini et al. (2018). First, flux random error was estimated as the variance of covariance, according to Finkelstein and Sims (2001):

$$\epsilon_{rand} = \frac{1}{N} \left( \sum_{j=-m}^{m} \hat{\gamma}_{w,w}(j)\hat{\gamma}_{c,c}(j) + \sum_{j=-m}^{m} \hat{\gamma}_{w,c}(j)\hat{\gamma}_{c,w}(j) \right)$$                  (7)



where $N$ is the number of datapoints (18 000 for 30 min of EC measurements at 10 Hz), $\hat{\gamma}_{w,w}$ is the variance and $\hat{\gamma}_{w,c}$ the covariance of the measured variables $w$ and $c$ (in this case, the vertical wind velocity and gas mixing ratio).

As the chosen processing schemes affect the resulting flux, the uncertainty related to the used processing options have to be accounted for. This uncertainty was estimated as

$$\epsilon_{proc} = \frac{max(F_{c,j}) - min(F_{c,j})}{\sqrt{12}} \tag{8}$$

where $F_{c,j}$ is the flux calculated according to $j=1,...,4$ different processing schemes: BA with 2D wind rotation, BA with planar fitting, LD with 2D wind rotation and LD with planar fitting. However, as all the different processing schemes will lead to slightly different random errors as well, the flux random error was estimated to be the mean of Eq. (7) for different processing schemes:

$$\overline{\epsilon_{rand}} = \frac{\sqrt{\sum_{j=1}^{4} \epsilon_{rand,j}^2}}{4} \tag{9}$$

The combined flux uncertainty is then the summation of $\epsilon_{rand}$ and $\epsilon_{proc}$ in quadrature:

$$\epsilon_{comb} = \sqrt{\overline{\epsilon_{rand}}^2 + \epsilon_{proc}^2} \tag{10}$$

To finally get the total uncertainty as the 95th percentile confidence interval, the total uncertainty becomes

$$\epsilon_{total} = 1.96\epsilon_{comb} \tag{11}$$

### 2.4.7 Advection

Horizontal and vertical advection are generally assumed negligible compared to EC and storage change fluxes, and advection is usually ignored as it is difficult to measure (Aubinet et al., 2010, 2012). However, advective transport may become important in low turbulence conditions, when turbulent exchange is restricted. Horizontal advection can only be determined from several EC towers placed in proximity, but this method comes with the uncertainty of natural spatial differences (Aubinet et al., 2005). Vertical advection, however, can be determined from the concentration profile as in Mammarella et al. (2007) and Lee (1998):

$$F_{VA} = \int_0^{z_r} \overline{w}(z) \frac{\partial \overline{c}(z)}{\partial z} dz = \overline{w}_{zr}(\overline{c}_{zr} - \langle \overline{c} \rangle) \tag{12}$$

where

$$\langle \overline{c} \rangle = z_r^{-1} \int_0^{z_r} \overline{c}(z) dz \tag{13}$$

where $\overline{w}$ is the mean vertical wind velocity, determined with the planar fitting method (Wilczak et al., 2001).





### 2.4.8 Flux gap-filling

Missing values of $CO_2$ were gap-filled according to Reichstein et al. (2005) while missing COS fluxes were replaced by simple
model estimates or by mean hourly fluxes in a way comparable to gap-filling of $CO_2$ fluxes (Reichstein et al., 2005).

    The COS gap-filling function was parameterised in a moving time window of 15 days to capture the seasonality of the fluxes.
To calculate gap-filled fluxes, the parameters were interpolated daily. Gaps where any driving variable of the regression model
was missing were filled with the mean hourly flux during the 15-day period.

We tested different combinations of linear or saturating (rectangular hyperbola) functions of the COS flux on photosynthetic
photon flux density (PPFD) and linear functions of the COS flux against VPD or RH. The saturating light response function
with fixed offset (=mean nighttime flux) explained the short-term variability of COS flux relatively well but the residuals as a
function of temperature, RH and VPD were clearly systematic. Therefore, for the final gap-filling, we used a combination of
saturating function on PPFD and linear function on VPD that showed good agreement with the measured fluxes while having
a relatively small number of parameters:

$$F_{COS} = a * I/(I + b) + c * D + d \tag{14}$$

where I is PPFD, D is VPD and a, b, c, and d are fitting parameters.

### 3 Results and Discussion

Here, the various processing schemes (see Fig. 1) are compared to a "standard scheme", consisting of linear detrending, $CO_2$
lag time, experimental high frequency correction and 2D wind rotation in the flux calculation. All results are presented in
Finnish winter time (UTC +2) and nighttime is defined as periods when PAR < 3 $\mu$mol m$^{-2}$s$^{-1}$.

### 3.1 Effect of lag time correction

Different lag time methods resulted in slightly different lag times and COS fluxes. As COS measurements are often char-
acterized by random noise, the maximization of the absolute value of cross-covariance may get stuck at local maxima, as
demonstrated in Fig. 2. The most common lag times were: 3.6 s from the COS lag method, 2.7 s from CO2 lag, 3.8 s from
RM lag (the lag window limit), and 2.7 s from the the DetLim lag (Fig. 3). Lag times determined from the COS and RM lag
methods were mostly at the upper limit of the lag window (3.8 s), whereas the lag from the $CO_2$ and DetLim methods reached
the global maxima around the window center.

    The lag time determined directly from $\overline{w'\chi'_{\mathrm{COS}}}$ resulted in a "mirroring effect" (Langford et al., 2015), i.e., fluxes close to
zero were not detected as often as with other methods, since the covariance is always maximized and the derived flux switches
between positive and negative values of similar magnitude (Fig. 4). The effect was not as strong with smoothed $\overline{w'\chi'_{\mathrm{COS}}}$ but
was still visible. Other methods had the PDF peak approximately at the same flux values but had otherwise small differences in
the distributions (Fig. 4). A constant lag time has been found to bias the flux calculation as the lag time likely varies over time





due to, for example, fluctuations in pumping speed (Langford et al., 2015; Taipale et al., 2010; Massman, 2000). A reduced

bias in the flux calculation with smoothed cross-covariance was introduced by Taipale et al. (2010), who recommended using
this method for any EC system with a low signal-to-noise ratio. However, we do not recommend this as a first choice since we
still find a mirroring effect with the RM lag in the final flux distributions. If other supporting measurements, like $CO_2$ in the
case of COS measurements, are not available to help the lag time determination, then RM lag would be a more suitable method
than the maximum cross-covariance method when determining lag times for fluxes with low signal-to-noise ratio.

About 40 % of the lag times were COS lag times with the DetLim lag method. Fig. 4 shows that the raw covariance of COS
only exceeds the noise level at higher COS flux values and thus the COS lag time is chosen by this method only at higher
fluxes, as expected. At lower flux rates, and especially close to zero, the COS fluxes are not high enough to surpass the noise
level and thus the $CO_2$ lag time is chosen.

The cumulative COS flux (summarized in Table 1) was highest when the lag time was determined from the detection limit

method (-74.3 nmol m$^{-2}$s$^{-1}$) and maximum $\overline{w'\chi'_{COS}}$ (-72.1 nmol m$^{-2}$s$^{-1}$). Using just the COS lag time biases against small
fluxes, as it is always maximizing the covariance, but it also produces more positive fluxes compared to the DetLim method.
This results in slightly smaller cumulative fluxes (smaller negative values). Maximizing the smoothed $\overline{w'\chi'_{COS}}$ produced the
third largest cumulative uptake (-69.7 nmol m$^{-2}$s$^{-1}$) while the $CO_2$ lag time produced a slightly higher cumulative flux (-65.9
nmol m$^{-2}$s$^{-1}$) than using a constant lag time of 2.7 s (-65.1 nmol m$^{-2}$s$^{-1}$). The difference between each of the cumulative

sums was 12 % or less. This difference is large in the annual scale, and we recommend using the detection limit method in lag
time determination of small fluxes, as in Nemitz et al. (2018).

**Table 1.** Cumulative sums of COS and $CO_2$ fluxes (in nmol m$^{-2}$s$^{-1}$ and mmol m$^{-2}$s$^{-1}$, respectively) and their difference to the reference
fluxes when using different processing options. Reference fluxes are indicated in bold.

| | Detrending | | | Lag time | | | | | Spectral corrections | | |
|---|---|---|---|---|---|---|---|---|---|---|---|
| | BA | LD | RF | COS lag | $CO_2$ lag | Const lag | RM lag | DetLim lag | Horst (1997) | Exp | None |
| COS | -66.0 | **-65.9** | -65.2 | -72.1 | **-65.9** | -65.1 | -69.7 | -74.3 | -61.2 | **-65.9** | -59.4 |
| Difference | 0.2 % | NA | 1.1 % | 9.4 % | NA | 1.2 % | 5.8 % | 12.7 % | 7.1 % | NA | 9.9 % |
| $CO_2$ | -12.71 | **-12.74** | -11.31 | NA | **-12.74** | NA | NA | NA | -12.54 | **-12.74** | -12.18 |
| Difference | 0.24 % | NA | 11.22 % | NA | NA | NA | NA | NA | 1.57 % | NA | 4.40 % |

## 3.2 Detrending methods

In order to check the contribution of different detrending methods on the resulting flux, we made flux calculations with different
methods: block averaging (BA), linear detrending (LD) and recursive filtering (RF) using the same lag time ($CO_2$ lag) for all

runs (Fig. 5). An Allan variance was determined for a time period when the instrument was sampling from a gas cylinder
(Werle, 2010). The time constant of 30 s for recursive filtering was determined from the Allan plot (Fig. 6), as the system starts
to drift in non-linear fashion at 30 s.

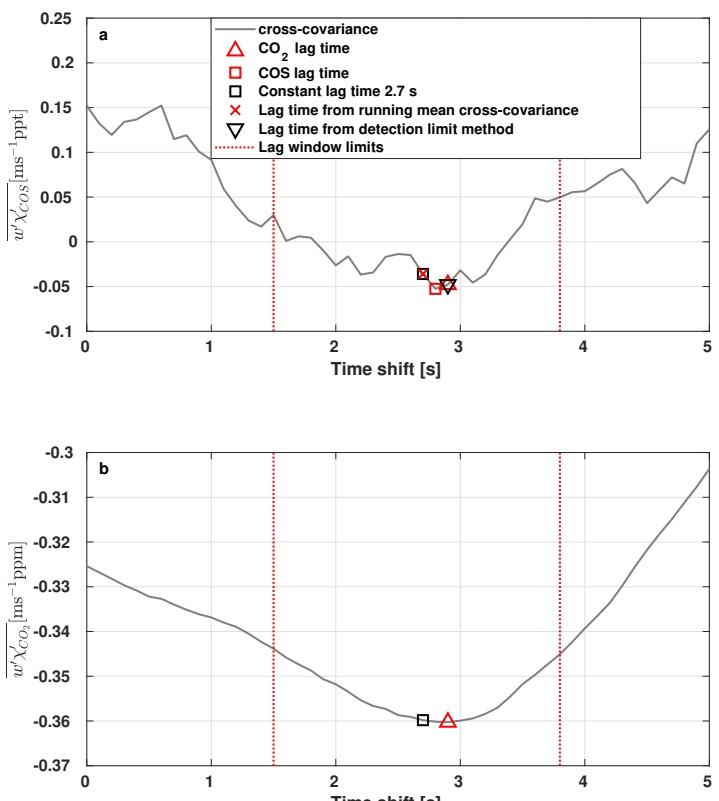

**Figure 2.** Lag time determined from different methods on 15 July 2015 at 12:00–12:30 for COS (a) and $CO_2$ (b).

The largest cumulative COS flux was obtained from BA (-66.0 nmol m$^{-2}$s$^{-1}$), since BA retains the fluctuations the most. The smallest cumulative sum resulted from RF (-65.2 nmol m$^{-2}$s$^{-1}$) while LD (-65.9 nmol m$^{-2}$s$^{-1}$) did not differ much
from BA (Table 1). The difference between BA and RF in the total cumulative COS flux was only 1.2 %. The variation of the COS flux was highest (from -234.2 to 154.9 pmol m$^{-2}$s$^{-1}$) when using the BA detrending method, consistent with a similar comparison in Gerdel et al. (2017). Variation for other detrending methods was from -119.1 to 79.7 pmol m$^{-2}$s$^{-1}$ for LD and from -75.3 to 33.3 pmol m$^{-2}$s$^{-1}$ for RF. While BA results in higher negative fluxes in general, it also includes higher positive fluxes, thus resulting in less variation in the cumulative sum.
For $CO_2$, the largest cumulative flux resulted from LD (-12.74 mmol m$^{-2}$s$^{-1}$) although the difference to BA (-12.71 mmol m$^{-2}$s$^{-1}$) was small. The smallest cumulative flux resulted from RF (-11.31 mmol m$^{-2}$s$^{-1}$), similar to COS. The difference between cumulative $CO_2$ flux with BA and LD was 11 %. The relative difference between RF and BA methods is thus smaller for COS than for $CO_2$, possibly because instrumental noise is aliasing part of the COS fluctuations.





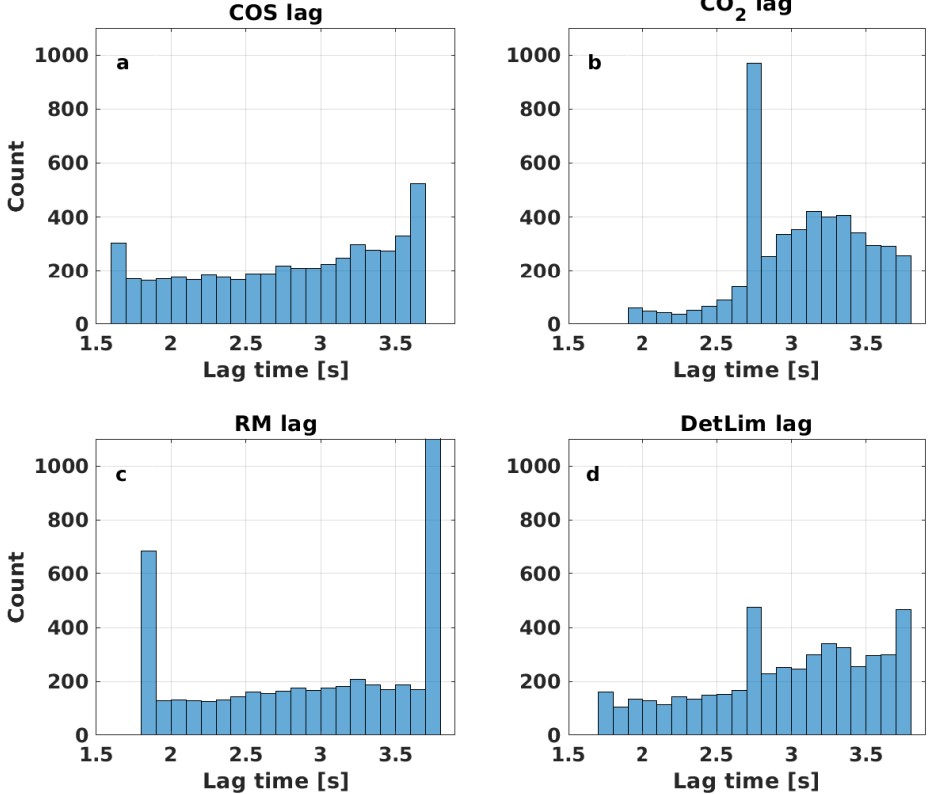

**Figure 3.** Distribution of lag times derived from different methods: COS lag (a), $CO_2$ lag (b), COS lag from a running mean cross-covariance (c) and combination of COS and $CO_2$ lag times (d), see Sec. 2.4.2.

The most commonly recommended averaging methods are BA (Sabbatini et al., 2018; Nemitz et al., 2018; Moncrieff et al.,
2004) and LD (Rannik and Vesala, 1999) because they have less impact on spectra (Rannik and Vesala, 1999) and require less
corrections in the low frequency. RF may underestimate the flux (Aubinet et al., 2012). As spectroscopic analyzers are prone
to e.g. fringe effects under field conditions, the use of RF might still be justified (Mammarella et al., 2010). Regular checks of
raw data provides information on instrumental drift and helps to determine the optimal detrending method. It is also recom-
mended to check the contribution of each detrending method on the final flux to better understand what is the low frequency
contribution in each measurement site and setup.





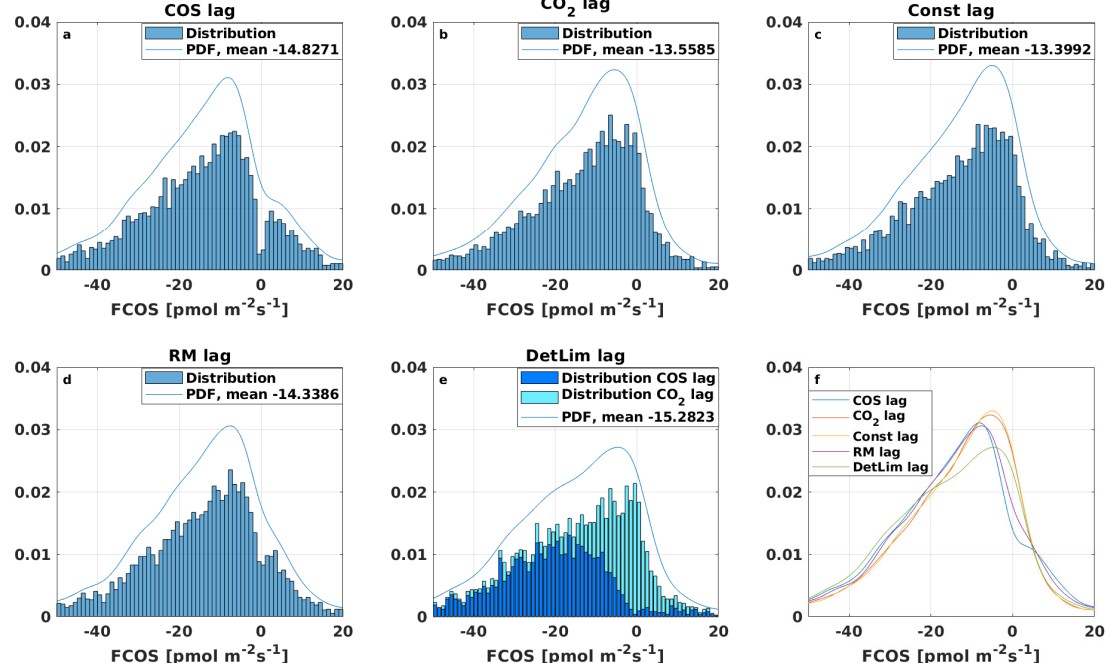

**Figure 4.** Normalized COS flux distributions using different lag time methods: COS lag (a), $CO_2$ lag (b), constant lag time of 2.7 s (c), COS lag from a running mean cross-covariance (d) and combination of COS and $CO_2$ lag times (e) (see Sec. 2.4.2), and a summary of all PDFs (f).

## 3.3 Frequency response correction

### 3.3.1 High frequency correction

The mean COS co-spectrum was close to the normal mean $CO_2$ co-spectrum (compare Fig. 7a and 7b). The power spectrum

of COS was dominated by white noise as can be seen from the increasing power spectrum with increasing frequency for normalized frequencies greater than 0.2, which is similar to what was measured for COS by Gerdel et al. (2017) and for $N_2O$ by Eugster et al. (2007). The fact that COS measurements are dominated by white noise at high frequencies means that those measurements are limited by the precision and that they do likely not capture the true variability in COS turbulence signals. This is less of a problem for $CO_2$, where white noise only starts to dominate at higher frequencies (3 s). Co-spectral attenuation

was found for both COS and $CO_2$ at high frequency. The response time of the analyzer (0.68 s) was calculated from $CO_2$ measurements and the same response time applied to COS high frequency spectral correction, assuming that both fluxes are affected by the same attenuation (Wehr et al., 2017).





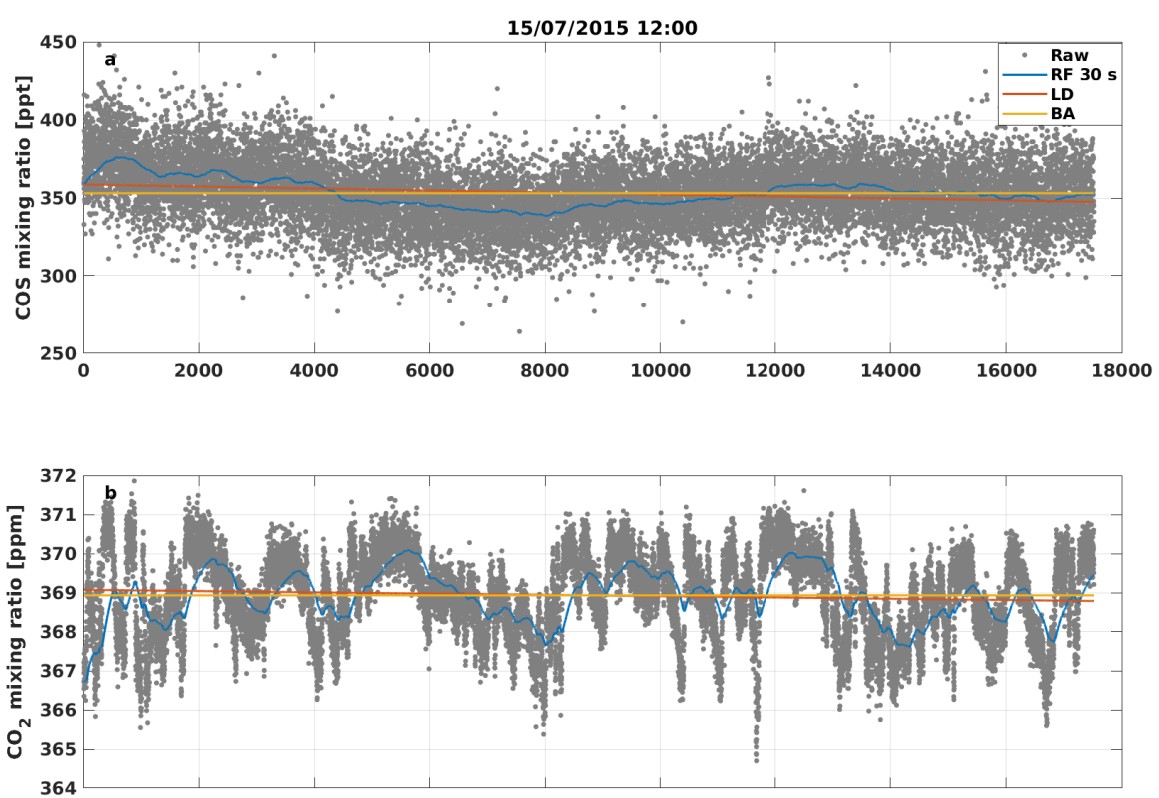

**Figure 5.** Comparison of different detrending methods (recursive filtering, linear detrending and block averaging, see Sec. 3.2) applied to raw COS (a) and $CO_2$ (b) mixing ratio data on 15 July 2015 12:00–12:30.

Both experimental and analytical approaches increased the COS and $CO_2$ fluxes, as expected. High frequency losses due to e.g. attenuation in sampling tubes and limited sensor response times are expected to decrease fluxes if not corrected for

(Aubinet et al., 2012). The cumulative sum of the COS flux, when using the $CO_2$ lag time and keeping low-frequency correction and quality filtering the same, was the lowest without any high-frequency correction (-59.4 nmol m$^{-2}$s$^{-1}$), highest with the experimental correction (-65.9 nmol m$^{-2}$s$^{-1}$) and in between with the analytical correction (-61.2 nmol m$^{-2}$s$^{-1}$). Correcting for the high frequency attenuation thus made a maximum of 9.9 % difference in the cumulative COS flux. Similar results were found for the $CO_2$ flux but the differences were smaller: without any high frequency correction the cumulative sum was lowest

at -12.18 mmol m$^{-2}$s$^{-1}$, highest with experimental correction (-12.74 mmol m$^{-2}$s$^{-1}$) and in between with the analytical correction (-12.54 mmol m$^{-2}$s$^{-1}$), thus making a maximum of 4.4 % difference in the cumulative $CO_2$ flux. Very similar



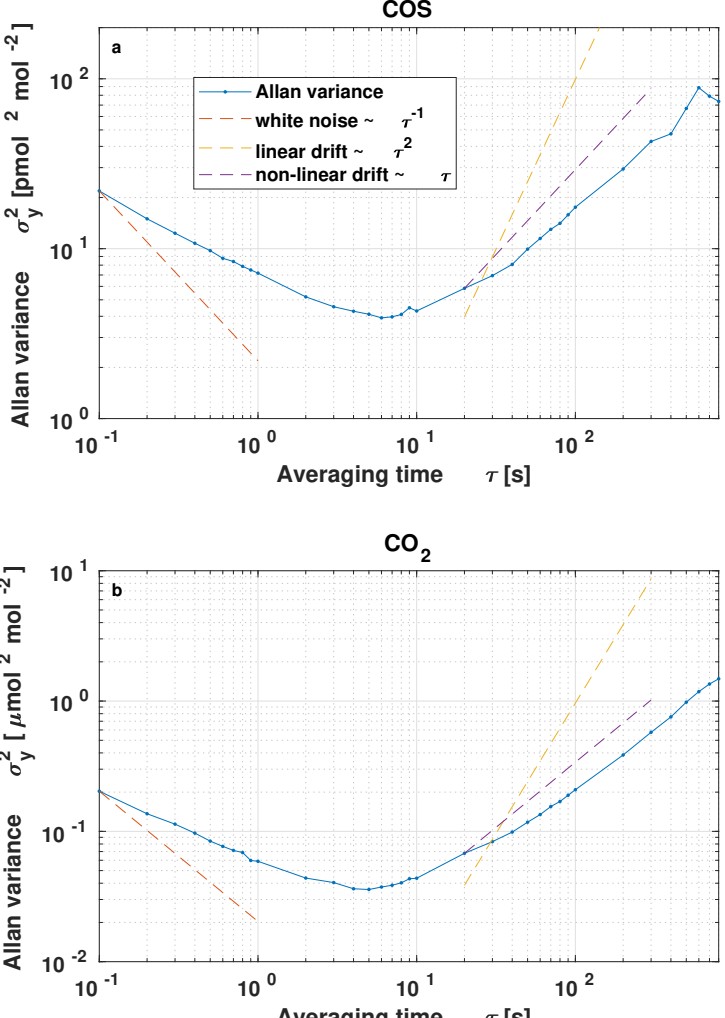

**Figure 6.** Allan plot for COS (a) and $CO_2$ (b) mixing ratios versus averaging time $\tau$. The dashed lines represent slopes for white noise, linear drifting and non-linear drifting.



results were found for $CH_4$ and $CO_2$ fluxes in Mammarella et al. (2016), where the high frequency correction made the largest difference in the final flux processing after dilution and Webb corrections.

From the empirical method we get a stability-dependent function for the cospectral peak frequency $n_m$ similar to the ana-
lytical method by Horst (1997) (Eq. 3 and 4):

$$n_m = 0.0956, \text{ for } \zeta \leq 0 \tag{15}$$

$$n_m = 0.0956(1 + 2.4163\zeta^{0.7033}), \text{ for } \zeta > 0 \tag{16}$$

This result is compared to the analytical method (Eq. 3 and 4) in Fig. 8. In contrast to Rannik et al. (2004), we find a difference between the empirical method (Eq. 16) and Horst (1997) (Eq. 4) in stable conditions (when $\zeta < 0$, Fig. 8).

### 3.3.2  Low frequency correction

As shown in Sect. 3.2, the difference between different detrending methods on the cumulative COS flux was negligible. From the example raw COS data (Fig. 5 a) it is also seen that the detrending methods are not very different from each other. Thus it is expected that the low frequency correction for COS does not make a large effect on the final flux, as noise is covering part of the low frequency variability in COS measurements. However, as we find larger differences in $CO_2$ detrending and the
detrending methods differ from each other more than for COS (Fig. 5 b), the low frequency correction is expected to be more relevant for $CO_2$ fluxes. Rannik and Vesala (1999) found a 15 % underestimation of the uncorrected fluxes for $CO_2$, which is also comparable to the difference between the $CO_2$ fluxes determined using different detrending methods in our study.

### 3.4  Storage change fluxes

In the following, storage change fluxes based on profile measurements are listed as default, with fluxes based on the constant
profile assumption listed in brackets.

The COS storage flux was negative from 15:00 in the afternoon until 06:00 in the morning with a minimum of -1.0 pmol m$^{-2}$s$^{-1}$ (-0.6 pmol m$^{-2}$s$^{-1}$) reached at 20:00 in the evening. A negative storage flux of COS indicates that there is a COS sink in the ecosystem when the boundary layer and effective mixing layer is shallow. Neglecting this effect would lead to overestimated uptake at the ecosystem level later when the air at the EC sampling height is better mixed. The COS storage flux
was positive from around 6:00 in the morning until 15:00 in the afternoon and peaked at 9:00 with a magnitude of 1.9 pmol m$^{-2}$s$^{-1}$ (0.8 pmol m$^{-2}$s$^{-1}$). The storage flux made the highest relative contribution to the sum of measured EC and storage fluxes at midnight with 18 % (13 %) (Fig. 9 c). The difference between the two methods was minimum 13 % at 11:00 and maximum 56 % at 09:00. The two methods made a maximum of 7 % difference on the resulting cumulative ecosystem flux, as already reported in Kooijmans et al. (2017).
The $CO_2$ storage flux was positive from 15:00 in the afternoon until around 4:00 in the morning with a maximum value of 0.62 $\mu$mol m$^{-2}$s$^{-1}$ (0.38 $\mu$mol m$^{-2}$s$^{-1}$) reached at 21:00 in the evening. A positive storage flux indicates that the ecosystem contains a source of carbon when the boundary layer is less turbulent and accumulates the respired $CO_2$ within the canopy. As



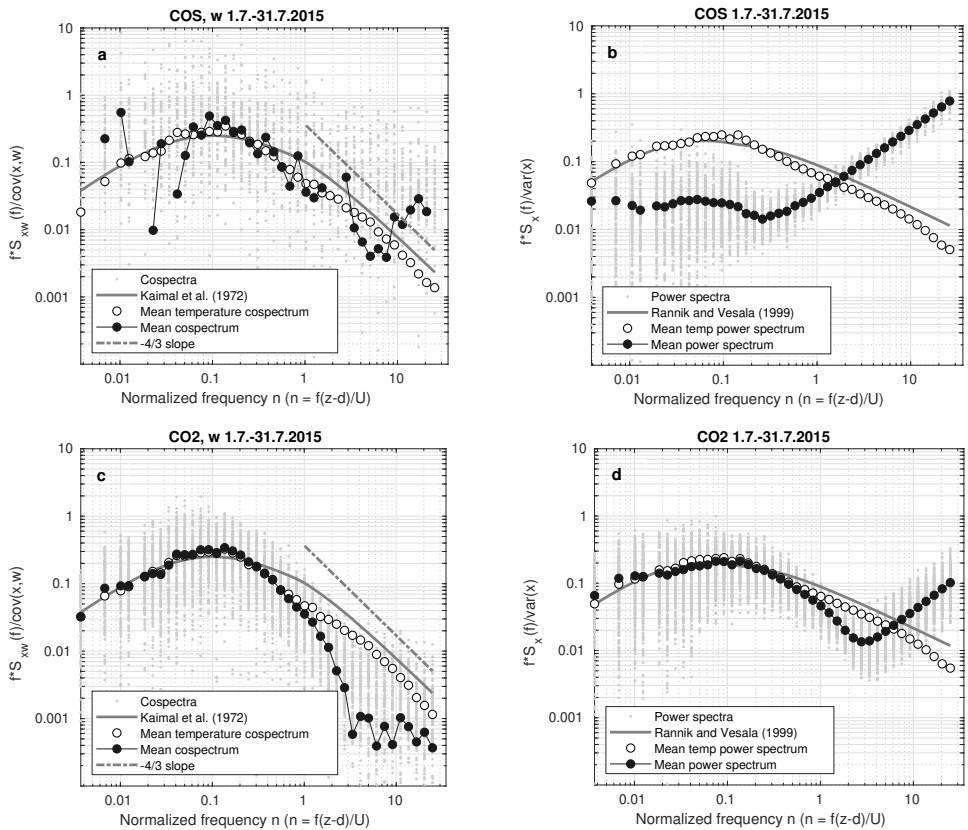

**Figure 7.** Co-spectrum and power spectrum for COS (a and b, respectively) and $CO_2$ (c and d) in July 2015.

turbulence would increase later in the morning, the accumulated $CO_2$ would result in an additional flux that could mask the gas

exchange processes occurring at that time step. The $CO_2$ storage flux minimum was reached with both methods at 08:00 with

magnitude -1.01 $\mu$mol m$^{-2}$s$^{-1}$ (-0.52 $\mu$mol m$^{-2}$s$^{-1}$) when the boundary layer has already started expanding, and leaves are

assimilating $CO_2$. The maximum contribution of the storage flux was as high as 89 % (36 %) compared to the EC flux at 18:00,

when the $CO_2$ exchange is turning to respiration and storage flux increases its relative importance (Fig. 9 d). The difference

between the two storage flux methods for $CO_2$ was maximum 53 % at 21:00 and minimum 13 % at midnight. The maximum

difference of 5 % was found in the cumulative ecosystem $CO_2$ flux when the different methods were used.

In conclusion, the storage fluxes are not relevant for budget calculations - as expected - but they are important to account for

the delayed capture of fluxes by the EC system under low turbulence conditions.

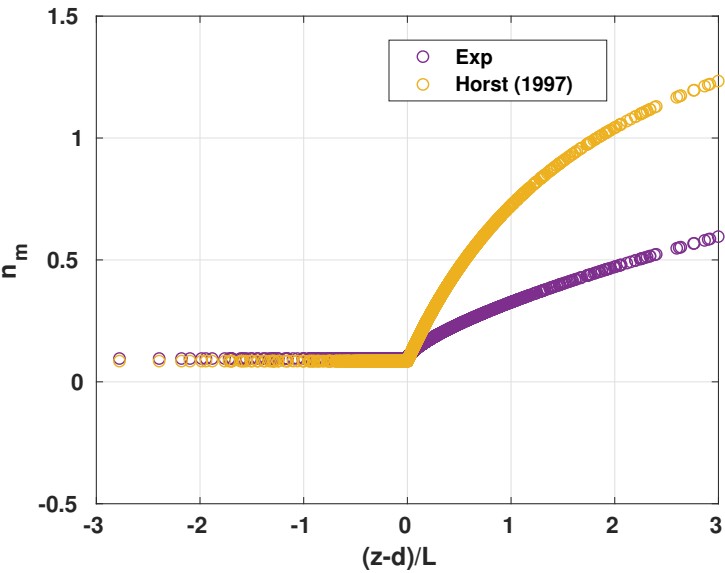

**Figure 8.** Co-spectral peak frequency as a function of stability according to experimental method and Horst (1997).

### 3.5 Vertical advection

Vertical advection of COS ($F_{VA,COS}$) was negligible during daytime and negative during nighttime (Fig. 10). During the night, $F_{VA,COS}$ was more important than the storage change flux and was half the size of the total ecosystem flux in magnitude.

Vertical advection of $CO_2$ ($F_{VA,CO2}$) was also negligible during the day, and in contrast to COS, it was slightly positive during night. $F_{VA,CO2}$ was still less than 50 % of the total ecosystem $CO_2$ flux and was comparable to the storage flux, similar to Mammarella et al. (2007) and Aubinet et al. (2003). Similar findings were reported also in Aubinet et al. (2005) at slightly sloping sites, for which the importance of storage change fluxes and advection varied with weather conditions. As we were not able to measure horizontal advection, we did not include vertical advection in the nighttime flux correction, to avoid an

overestimated correction and an underestimated total flux (Aubinet et al., 2003). Horizontal and vertical advection were found to be large but of opposite signs in Aubinet et al. (2003) and they suggested to exclude the correction based solely on vertical advection, proposed by Lee (1998).

### 3.6 $u_*$ filtering

Especially during nighttime it is common that there is not enough turbulence to mix the surface layer. In this case, storage and

advection start to play a bigger role and the measured EC flux of a gas does not reflect the atmosphere-biosphere exchange, typically underestimating the exchange. This often leads to a systematic bias in the annual flux budgets (Moncrieff et al., 1996;



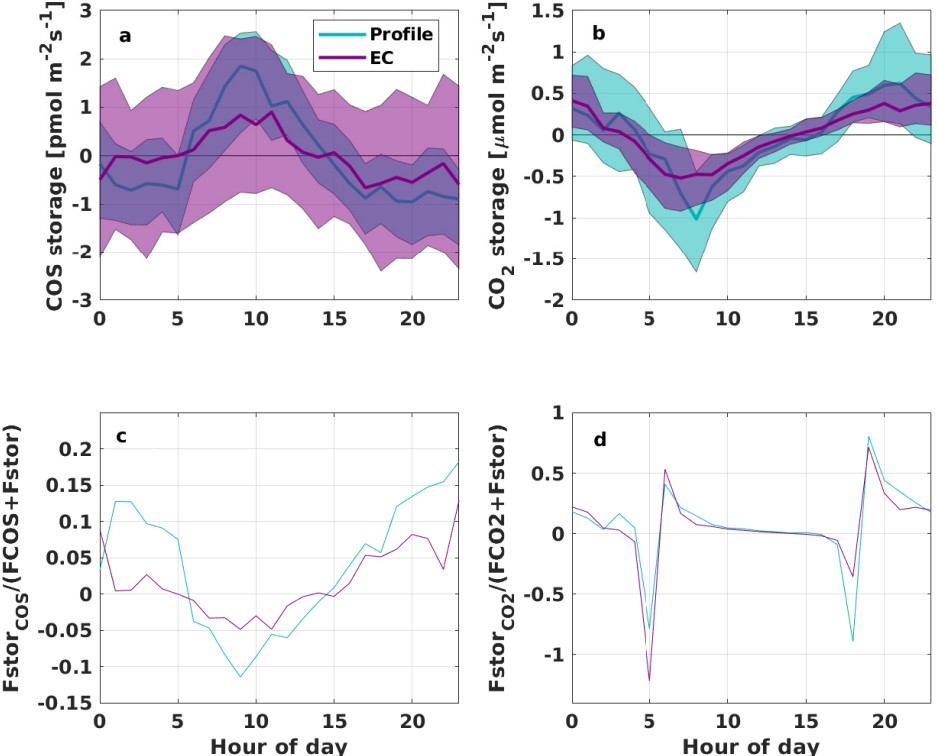

**Figure 9.** Diurnal variation of the storage change flux, determined from COS (a) and $CO_2$ (b) profile measurements (blue) and by assuming a constant profile up to 23 m height (purple). Contribution of storage change flux to the total ecosystem EC flux with the profile measurements and assuming a constant profile for COS (c) and $CO_2$ (d).

Aubinet et al., 2000; Aubinet, 2008). Even after studies of horizontal and vertical advection, the $u_*$ filtering still keeps its place as the most efficient and reliable tool to filter out erroneous data under low turbulence (Aubinet et al., 2010).

For COS, the nighttime filtering is a more complex issue than it is for $CO_2$. In contrast to $CO_2$, COS is taken up by the ecosystem during nighttime (Kooijmans et al., 2017, 2019), depending on stomatal conductance and the concentration gradient between the leaf and the atmosphere. When the atmospheric COS concentration goes down under low turbulence conditions (due to nighttime COS uptake in the ecosystem), the concentration gradient between the leaf and the atmosphere goes down, such that a decrease in COS uptake can be expected (Kooijmans et al., 2017). The $u_*$ filtering is applied to conform the assumption that fluxes do not go down under low turbulence conditions, as is the case for respiration of $CO_2$, but which does

not apply to COS uptake. The $u_*$ filtering may therefore bias COS fluxes due to false assumptions. Still, the $u_*$ filtering is applied here to overcome the EC measurement limitations under low turbulence conditions.





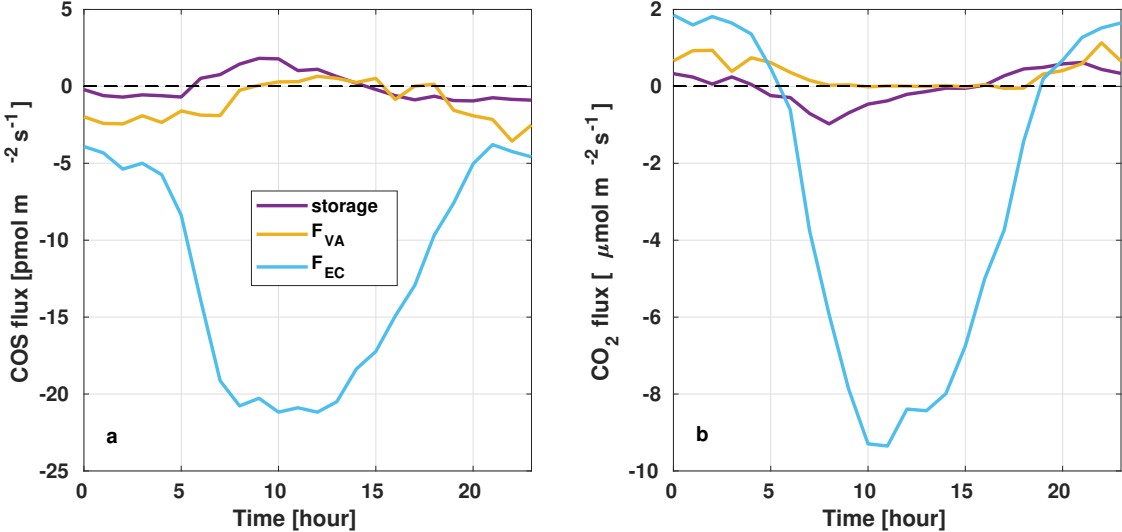

**Figure 10.** Diurnal variation of storage change flux (purple), vertical advection (yellow) and total ecosystem EC flux (blue) for COS (a) and $CO_2$ (b).

We determined u* limits of 0.23 ms$^{-1}$ for COS and 0.22 ms$^{-1}$ for $CO_2$ (Fig. 11). Filtering according to these $u_*$ values would remove 12 % and 11 % of data, respectively. If the storage change flux was excluded when determining the $u_*$ threshold, the limits were 0.39 ms$^{-1}$ and 0.24 ms$^{-1}$ from $CO_2$ and COS fluxes, respectively. The increase in the $u_*$ threshold with $CO_2$
is because the fractional storage flux is larger for $CO_2$ than for COS (Fig. 9, panels c and d). On the other hand, the $u_*$ limit for COS stayed similar to the previous one. With these $u_*$ thresholds the filtering would exclude 30 % and 13 % of the data for COS and $CO_2$ respectively.

If fluxes are not corrected for storage before deriving the $u_*$ threshold, there is a risk of overestimation due to double accounting. Without storage correction done first, the fluxes during low turbulence in nighttime would be removed and storage
flux ignored. The filtered flux would be gapfilled, thereby accounting for storage by the canopy, but then accounted for again when the storage is released and measured by the EC system in the morning (Papale et al., 2006). Thus, it is necessary to make the storage flux correction before deriving $u_*$ thresholds and applying the filtering.

## 3.7 Gap-filling

For COS fluxes, 44 % of daytime flux measurements were discarded due to the above-mentioned quality criteria (Sect. 2.4.4) and low-turbulence filtering. As expected, more data (66 %) were discarded during nighttime. Altogether 52 % of all COS flux data were discarded and gap-filled with the gap-filling function presented in Eq. (14). The cumulative sum of the final,

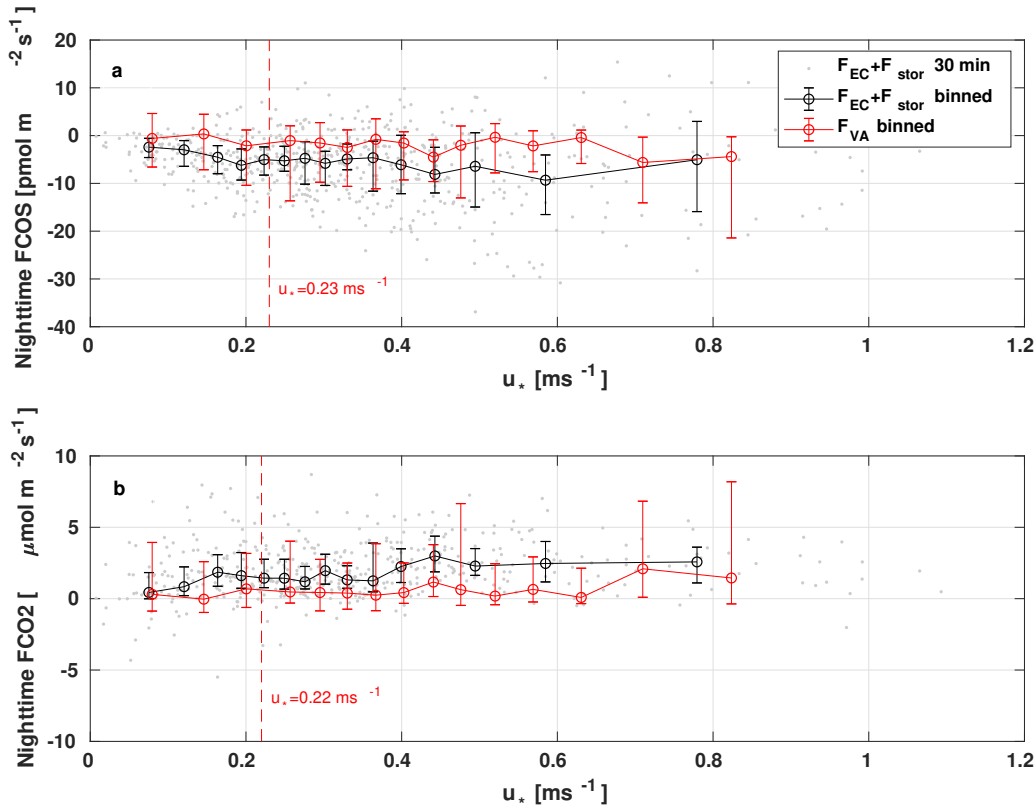

**Figure 11.** Nighttime COS (a) and $CO_2$ (b) fluxes binned to 15 equal-sized friction velocity bins. Friction velocity thresholds are shown with the red dashed lines.

corrected and gap-filled, COS fluxes during the whole measurement period totalled up to -139 $\mu$mol COS m$^{-2}$, while without gap-filling the cumulative sum would be 47 % smaller at -73.7 $\mu$mol COS m$^{-2}$.

For $CO_2$, 41 % daytime $CO_2$ fluxes were discarded, while 67 % of fluxes were discarded during nighttime. Altogether comprising 53 % of all $CO_2$ flux data. $CO_2$ fluxes were gap-filled according to standard gap-filling procedures presented in (Reichstein et al., 2005). The cumulative NEE after all corrections and gap-filling was -22.1 mol $CO_2$ m$^{-2}$, while without gap-filling the cumulative sum would be only -19.5 mol $CO_2$ m$^{-2}$.

**3.8   Errors and uncertainties**

The uncertainty due to different processing schemes in the flux processing contributed 33 % to the total uncertainty of the COS flux, the rest was composed of the random flux uncertainty (Fig. 12). For the $CO_2$ flux uncertainty, the processing was more





important than for COS (40 %), but the random error was still dominating the combined flux uncertainty. The random error of the $CO_2$ flux was found to be lower than in Rannik et al. (2016) for the same site, probably related to differences in the gas

analyzers and overall setup. The mean noise estimated from Lenschow, D. and Wulfmeyer, V. (2000) was 0.06 $\mu$mol m$^{-2}$s$^{-1}$ for our QCLS $CO_2$ fluxes while in Rannik et al. (2016) it was approximately 0.08 $\mu$mol m$^{-2}$s$^{-1}$ for LI-6262 $CO_2$ fluxes at the same site. Gerdel et al. (2017) found the total random uncertainty of COS fluxes to be mostly around 3–8 pmol m$^{-2}$s$^{-1}$, comparable to our results.

The relative flux uncertainty for COS is very high at low flux (-3 pmol m$^{-2}$s$^{-1}$ < FCOS < 3 pmol m$^{-2}$s$^{-1}$) values (8 times

the actual flux value) but levels out to 45 % at fluxes higher (meaning more negative fluxes) than -27 pmol m$^{-2}$s$^{-1}$ (Fig. 12c). The total uncertainty of the $CO_2$ flux was also high at low fluxes (-1.5 $\mu$mol m$^{-2}$s$^{-1}$ < FCO2 < 1 $\mu$mol m$^{-2}$s$^{-1}$, uncertainty reaching 120 % of the flux at 0.17 $\mu$mol m$^{-2}$s$^{-1}$) and decreased to 15 % at fluxes higher than -11 $\mu$mol m$^{-2}$s$^{-1}$ (Fig. 12d). Higher relative uncertainty at low flux levels is probably due to detection limit of the measurement system.

## 4   Conclusions

In this study, we examined the effects of different processing steps on COS EC fluxes and compared them to $CO_2$ flux processing. COS fluxes were calculated with five lag time determination methods, three detrending methods, two high frequency spectral correction methods and with no spectral corrections. We calculated the storage change fluxes of COS and $CO_2$ from two different concentration profiles and investigated the diurnal variation in the storage change fluxes and vertical advection. We also applied $u_*$ filtering and introduced a gap-filling method for COS fluxes. We also quantified the uncertainties of COS

and $CO_2$ fluxes.

The largest differences in the final fluxes came from lag time determination and spectral corrections. Different lag time methods made a difference of maximum 12.7 % in the cumulative COS flux while spectral corrections influenced the cumulative flux by 9.9 %. We suggest to use a combination of COS and $CO_2$ lag times for COS flux calculation, depending on the COS flux uncertainty, so that potential biases in the determined lag times for small fluxes can be eliminated. Experimental high

frequency correction is recommended for accurately correcting for site specific spectral losses. Different detrending methods made a maximum of 1.1 % difference in the cumulative COS flux while it was more important for $CO_2$ (11.22 % difference between linear detrending and recursive filtering). We recommend comparing the effect of different detrending methods on the final flux for each site separately, to determine the site and instrument specific trends in the raw data.

Flux uncertainties of COS and $CO_2$ followed a similar trend of higher relative uncertainty at low flux values and random flux

uncertainty dominating over uncertainty related to processing in the total flux uncertainty. The relative uncertainty was more than 5 times higher for COS than for $CO_2$ at low flux values (absolute COS flux less than 3 pmol m$^{-2}$ s$^{-1}$), while at higher fluxes they were more similar.

We emphasize the importance of lag time method selection for small fluxes, whose uncertainty may exceed the flux itself, to avoid systematic biases. COS EC flux processing follows similar steps as other fluxes with low signal-to-noise ratio, such as

$CH_4$ and $N_2O$, but as there are no sudden bursts of COS expected and its diurnal behaviour is close to $CO_2$, some processing



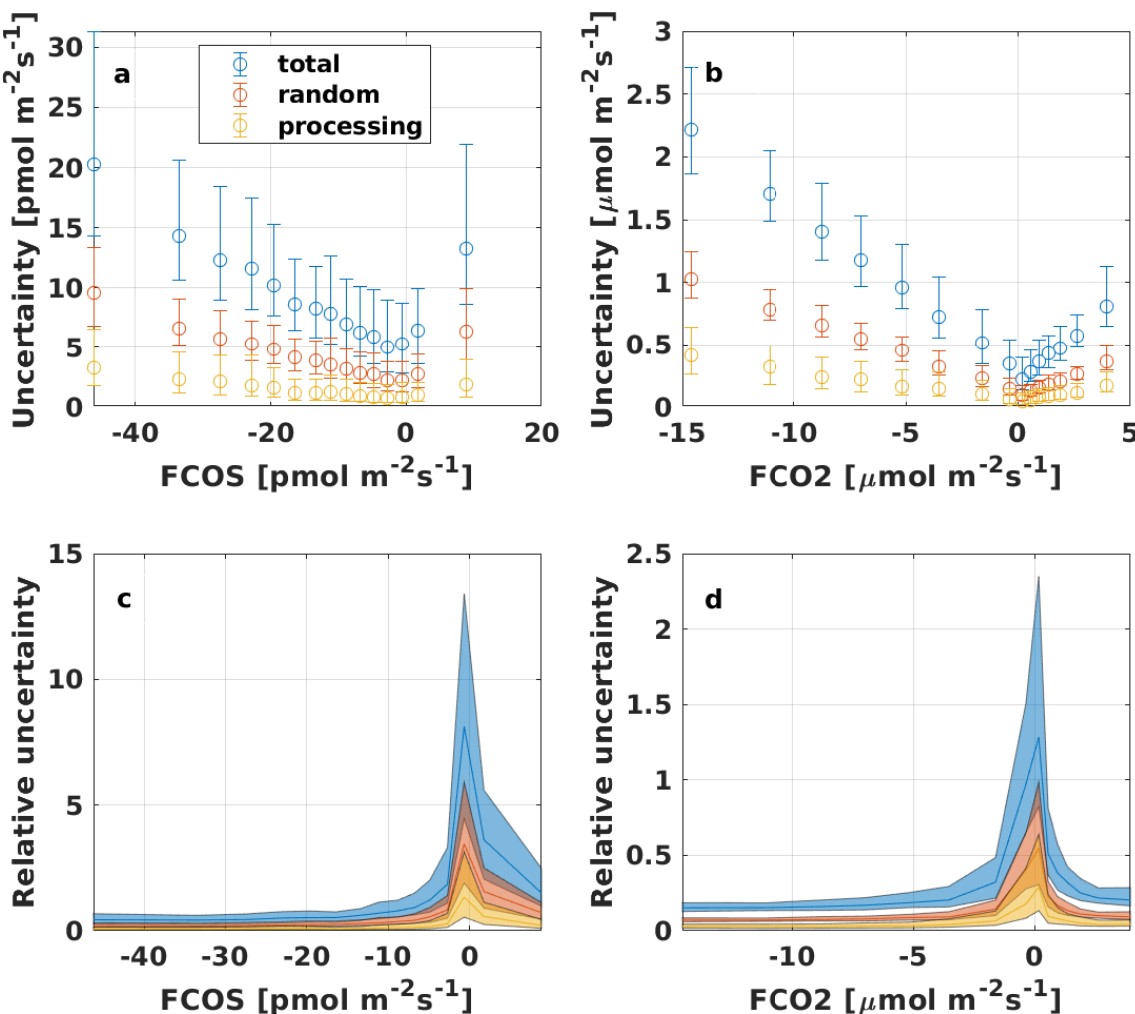

**Figure 12.** Uncertainty of COS and $CO_2$ fluxes, binned to 15 equal-sized bins that represent median values (a,b). Error bars show the 25th and 75th percentiles. Total uncertainty is represented as the 95 % confidence interval (1.96 $\epsilon_{comb}$) Panels c and d represent the relative uncertainty, i.e. the uncertainty divided by the flux, for COS and $CO_2$, respectively.



steps are more similar to $CO_2$ flux processing. In particular, lag time determination and high-frequency spectral corrections should follow the protocol of low signal-to-noise ratio fluxes (Nemitz et al., 2018) while QA/QC, despiking, $u_*$ filtering and storage correction should follow the protocol produced for $CO_2$ flux measurements (Sabbatini et al., 2018).

*Data availability.* Data sets will be open and available before the final submission.

*Author contributions.* KME and IM designed the study and KME processed and analyzed the data. PK developed the gap-filling function for COS and participated in data processing. IM, LK, US and HC participated in field measurements. KME and IM wrote the manuscript with contributions from all co-authors.

*Competing interests.* The authors declare that they have no conflict of interest.

*Acknowledgements.* We thank the Hyytiälä Forestry Field Station staff for all their technical support, especially Helmi-Marja Keskinen
and Janne Levula. KME thanks The Vilho, Yrjö and Kalle Väisälä foundation for their kind support. The authors thank ICOS-FINLAND (319871), the Academy of Finland Center of Excellence (307331) and Academy Professor projects 284701 and 282842, and the ERC-advanced funding scheme (AdG 2016 Project number: 742798, Project Acronym: COS-OCS).



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
