# Peer review of "Towards standardized processing of eddy covariance flux measurements of carbonyl sulfide"

_Atmospheric Measurement Techniques, 2019_

## Referee Comment (RC1) · Georg Wohlfahrt (Referee) · 31 Oct 2019

General assessment: Kohonen et al. report on the effects of varying various post-processing steps required for eddy covariance COS flux measurements with the aim, as stated in the title, to standardize these. COS EC flux measurements are increasingly making their way into the literature as COS offers a novel means of constraining GPP and stomatal conductance. Yet, the necessary processing steps are way not as harmonized as is the case for $CO_2$, potentially causing systematic bias between studies using different processing schemes, thus impeding synthesis activities. Overall I think this is a timely and relevant addition to the literature, which fits with the scope of

the journal. I though also believe that the manuscript suffers from several issues, which will require significant changes, as detailed below.

Major comments: (1) First, I have several formal issues with the manuscript: English style is often poor, which creates situations in which the intended meaning is not entirely clear (e.g. l. 32 the explanation of footprint limitations during stable stratification). Some of the formulations are too sloppy and thus misleading (e.g. l. 33 where "operation at high frequency" is mixed with "fast time response"). Some text is trivial or circular (e.g. l. 434-435), some of the concepts are wrong (e.g. l. 49) and some information is missing (e.g. legend of Fig. 11). Often some later, in-house studies are cited instead of the original papers. Finally, a mix of tenses is used when typically the past tense should be used to describe own results.

(2) Novelty and justification of the study: In 2017 a methodological paper on COS EC flux measurement post-processing was published in the same journal (Gerdel et al.). The authors justify their paper mainly by stating that their analysis goes beyond this previous paper. While this is partially true (in particular the analyses on lag times is novel), I think the authors should follow what the title of the paper suggests and rather sell their work as contributing towards a standardization of COS EC flux post-processing routines. To this end, I suggest to synthesize, e.g. in a table, the various processing steps that were used by previously published studies as a starting point and use this as a backbone for their analysis and the resulting recommendations. This table would then summarize whether and if so how previous studies detrended their time series, how the lag time was found, how low/high-frequency response corrections were applied, whether data were filtered for low u* (how were thresholds found) and which QC/QA was used. Following this suggestion requires at least the introductory section to be more or less completely re-written and would allow the paper to live up to what its title suggests and eventually become a reference for COS EC flux measurements.

(3) Vertical advection: This section is somewhat odd – the authors acknowledge that knowing the magnitude of vertical advection is meaningless unless the magnitude of

horizontal advection is known as well, yet vertical advection is reported even though horizontal advection has not been quantified. Unless the authors can come up with a discussion of what their results on the sign and magnitude of vertical advection actually mean in the context of their study, I thus suggest removing the results on vertical advection and all text/material that pertains to it.

(4) Corrections for high-frequency flux loss: Comparing two different approaches is novel for COS, yet surprisingly none of the underlying results are shown – I suggest to expand this section.

(5) Changes of co-spectral peak frequency with stability: Among the results of this study is a figure comparing the changes in the co-spectral peak frequency with stability for the Horst model and this study. While interesting, this analysis and the results are not motivated in the introduction and are barely discussed. Again, unless the authors are able to come up with a discussion of what the observed differences mean for their study, I suggest removing this material (or possibly moving it into a supplement).

(6) Gap-filling: This is an indeed novel aspect, however way underexploited by the authors. Only a single arbitrarily chosen gap-filling algorithm is tested, the authors miss to put it to a true test and results of gap-filling (e.g. time course of estimated parameters and selected results illustrating gap-filling behaviour) are lacking.

(7) Li-6262 and QCL CO2 cross-covariance maximisation: An important detail of this study is the setup, which includes a closed-path IRGA that is used to measure CO2 and H2O concentrations (not clear whether from the same tube as the QCL). These data are used to account for the drift in the computer clocks acquiring the sonic (& IRGA) and QCL data, respectively. While this nicely shows the benefit of having a complementary suite of measurements at "super-sites" such as Hyytiälä, in my view the reliance on an additional instrument is a drawback of this study as it limits the applicability of the proposed approach at other sites where no additional IRGA or an open-path model or closed-path model with a short tube is deployed. Even more so,

this approach is unnecessary, as there are simpler, software-based, solutions available to keep computer clocks in a network synchronized. In addition, by aligning data this way, the authors ruin a truly independent means of cross-comparing the QCL CO2 and H2O fluxes. I thus think it would be useful to explore the possibility of aligning the data sets in time without the help of the IRGA data. This is possible by expanding the time window in which the lag determination algorithm searches for, as with computer clocks being reset only once a day, time shifts of several seconds may result (in both directions). The reliability of this approach may then be checked by comparing against lag times and fluxes calculated with the IRGA.

(8) Conclusions: The authors should conclude with referencing against what processing steps have been used by previous studies and put their results into perspective with these, by highlighting critical steps and the need for further harmonization.

Detailed comments: l. 1-3: reformulate to better convey intended meaning "... growing in popularity with the aim of estimating gross primary productivity at ecosystem scale, however lack standardized protocols ..." l. 17: replace "due to the use" with something more suitable, e.g. "motivated by ..." l. 20: ", but in contrast to CO2, COS is destroyed ..." l. 22: for readers not familiar with the LRU, talking about the radiation-dependency of the LRU without introducing the concept might be highly confusing l. 29: "... the assumptions underlying the EC method ..." l. 29-34 and l. 35-41: these two paragraphs are in my view too general to meaningfully add to the introduction l. 32: reformulate – what you likely mean is that the measurement height should be such that the footprint remains within the ecosystem of interest even during stable stratification l. 33: EC instruments need to have a fast time response, which is different from "operation at high frequency" (a slow-response sensor does not become suitable for EC only because its data are logged at 20 Hz), as it can be shown that fast-response measurements made every few seconds do not cause a systematic bias in the EC flux l. 38: not sure this sentence applies universally to all closed-path analyzers and anyway I would think this is too much detail for the introduction – suggest to remove l. 41: the first ones to report on

this issue were Ibrom et al. (2007) l. 43-44: a rotation into the prevailing wind direction is only one step in the coordinate rotation; typically the aim is to align the coordinate system with the prevailing streamlines (2D or 3D) or with respect to some coordinate system that was established over a longer period (e.g. planar fit) l. 49: the EC flux is fine – the problem is that it may represent a poor estimate of the surface-atmosphere exchange under these conditions l. 53: in fact it was the following year (1999) that John Finnigan published a commentary on the Lee (1998) paper in which he demonstrated that correcting only for vertical advection is nonsense l. 56: if environmental data are lacking too, mean diurnal variation may be used as a last resort l. 64: if the cross-covariance is flat, then a wrong lag time will not have a large effect l. 66: I do not get the "However, . . ." which links to the previous sentence, which does not appear to make sense here l. 69: Gerdel et al. did study lag determination (their section 3.1) and u*-filtering (their Fig. 6) l. 71: actually you do not discuss the "EC flux measurement setup" at all l. 71-73: the introduction should finish with a statement of objectives l. 79: coordinate rotation l. 78-88: does this have any relevance for this study? l. 92: and sonic temperature l. 95: flow rate through Li-6262, same pump as QCL, tube diameter, length, is the same tube as for the QCL? l. 106: is this the mean? what is the standard deviation? l. 111-112: given the precision of typical computer clocks, this will result in clock differences up to several seconds; important to add that most likely the Li-6262 data were acquired by and thus synchronized with the sonic anemometer through it's A/D input?! l. 129-131: I am not sure I understand the first step – the QCL clock may be either delayed or advanced with respect to the clock of the sonic anemometer & IRGA and I thus do not understand why you shift the QCL time series by the lag time between w and IRGA CO2? In my understanding you could start off with the second step which actually aligns both time series. l. 131: might be worth showing this as a histogram in the supplement? l. 133-134: shouldn't this sentence come first in this section as this likely was the initial step? Or did the procedure described in l. 126-132 use data before despiking? l. 142-143: not sure that computation time is a relevant issue nowadays with regard to coordinate rotation Fig. 1: the first rotation angle is typically the one that

aligns the coordinate system along the main wind direction – why would that rotation be limited to less than 10°, which would mean rejecting fluxes from 340°? l. 200: in my memory, the first to propose this approach were Aubinet et al. (2000, 2001) l. 201: a site-specific cospectral model was already used by Wohlfahrt et al. (2005) l. 213: which rotation angle? l. 232: but fluxes are available half-hourly – how do you come up with a half-hourly storage change estimate? l. 237-238: 5 hours sounds like a really long time to reduce random noise – in fact I would expect a 5 hour moving average to even average out true storage; how exactly did you calculate the one-point storage term? l. 264: clearly here only Lee (1998) is to credit with this approach l. 270: "missing CO2 fluxes"? l. 282: add units l. 284-286: isn't this a repetition from above? l. 288-289: all measurements are characterised by noise to a certain degree ... l. 289-290: I do see several local minima in Fig. 2, but not that any of the tested algorithms gets stuck in one of these l. 298-299: so what? How does that sentence relate to your results? l. 305: reformulate – I guess that what you mean is that with the DetLim method, the COS lag was selected in 40 % of all cases (while the CO2 lag was chosen in 60 %) l. 305: can you compare lag times of COS and CO2 both fluxes are clearly higher than the flux detection limit? Is there a systematic difference between the two (e.g. COS lag time always longer than CO2) and if so could the DetLim Method be improved by adding this offset to the CO2 lag instead of just using the CO2 lag? l. 309: why did you choose the cumulative flux as a metric? Wouldn't the cumulative flux potentially be affected by compensating effects (over- and underestimation during certain conditions resulting in similar cumulative fluxes)? Unlike for CO2, I also do not see much need to calculate a daily or longerterm budget for COS; I also do not see how you can get units of nmol/m2s for COS as for a cumulative flux you need to integrate over time, i.e. multiply the half-hourly flux by 1800 s and then sum these up this then yields molar units per m2 and time period over which the cumulative was calculated – same for CO2. Table 1: instead of repeating the values for the reference three times, list it only once?! l. 315: how do you know this difference is large on an annual scale? What is the basis for this recommendation? l. 335-336: "low-frequency corrections" l. 336-337:

[Figure]

combine both sentences l. 337-338: this is kind of trivial l. 344: what is the "normal" CO2 cospectrum? Fig. 7a shows the COS cospectrum, 7b the COS power spectrum – this sentence does not make sense Fig. 7: are the (co)spectra in any way filtered for stability or really averaged over the entire month? I suggest to remove the sub-grid lines in all four panels as otherwise these are too busy and become blurry l. 349: in Fig. 7 you are using normalized frequency, thus no units l. 350: while the attenuation is clearly visible for CO2, it does not show well for COS, whose cospectrum mostly overlaps with the sensible heat cospectrum l. 350-351: how was this calculated? l. 353: indeed this is as expected . . . remove? l. 355-357: I think it would be instructive to show at least one characteristic example comparing the experimental and analytical frequency response correction approaches as otherwise this remains a "black box" for the reader l. 358: while I understand that the experimental frequency response correction approach is part of the standard against which the comparison is made, I think (i) that the no frequency response correction scenario is useless as we know that this leads to a bias and (ii) instead it would be useful to compare the magnitude of the correction between the analytical and experimental approach in order to understand how much of a difference it makes whether one or the other correction is used l. 364-369: this section comes a bit as a surprise as it never has been mentioned as a goal before to do this kind of comparison and also lacks a proper discussion – without discussion I rather suggest to remove Fig. 8 and the corresponding text; one point of discussion might be how much the difference in the cospectral reference models contributes to the differences between the analytical and experimental approach; btw., the results in Fig. 8 are qualitatively consistent with Fig. 11 of Wohlfahrt et al. (2005) l. 371-377: here I have the impression that you not describing the actual magnitude of the correction, but rather what we might expect based on one example shown in Fig. 5?! l. 372-374: this expectation would only be justified if the algorithm used for correcting for flux loss at lower frequencies would "know" of the noise, which I expect it does not Figure 9: to what period do the data shown refer to? l. 400-401: again, this is well established - l. 418: I would not call data erroneous – they simply do not fulfil the assumptions

underlying our simplified model of surface-atmosphere exchange l. 421-423: with this reasoning, wouldn't it make sense to get rid of the concentration dependency by using the deposition velocity, i.e. the flux normalized with the concentration, instead of the flux itself? Or perform the analysis with data stratified by COS concentration to minimize the issue? l. 428-437: knowing that the u*-filtering needs to be applied on the sum of the storage and EC flux, why do you still give the numbers for the EC flux without storage? l. 434-435: circular argument - without applying the storage correction it is pretty clear that the storage flux is clear that the storage flux is ignored Figure 11: also shows the vertical advection without being mentioned in the figure legend?! l. 442-444: is this a useful comparison? The cumulative COS flux must be less negative if missing data (which are generally negative) are not gap-filled! In order to evaluate the skill of the gap-filling algorithm the authors need to create artificial (but realistic) gaps in their time series (see CO2 flux literature to that end) and then compare the gap-filled against the measured (during the artificially created gaps) fluxes! l. 474-475: why do you recommend the experimental high-frequency correction approach? Is it more accurate? If so this needs to be demonstrated! What about the performance of the experimental approach in situations with low/noisy sensible heat cospectra and what about the effect of the QCL on the ratio central to the experimental approach?

---

## Referee Comment (RC2) · Anonymous Referee #3 · 5 Nov 2019

Overview

Erkkilä (Kohonen) et al present a detailed and valuable analysis of the impact of various eddy covariance data processing options on the calculated ecosystem uptake of carbonyl sulfide (COS). They attempt to quantify the flux uncertainty deriving from the data processing, and they make recommendation for some of the options. The methods are sound and the manuscript is fairly well written and easy to follow.

I do have some concerns about the analysis and interpretation of the results. In addition to the "major comments" of referee Wohlfahrt, with which I agree, I believe the paper could be made stronger by addressing the issues below.

[Figure]

Specific Scientific Comments

1. I disagree with the idea that the "processing uncertainty" reported here is actually an uncertainty in the calculated fluxes. Instead, it is a metric of the sensitivity of the calculated fluxes to different processing choices. Some of those choices are clearly better than others, and it doesn't make sense to calculate the fluxes in a way that is known to be pretty good and in another way that is known to be pretty bad and then say that the difference between the two ways is the uncertainty in the flux. In particular, the following data processing choices are obviously bad: (a) the COS lag and RM lag methods, (b) the RF 30s detrending method, (c) omitting high-frequency correction, (d) omitting the storage flux, (e) determining a $u^*$ filter threshold before including the storage flux, (f) omitting gap filling (for cumulative sums). None of those methods should be included when assessing methodological uncertainty, as there is no uncertainty about the fact that those methods should not be used. Thus the "processing uncertainties" presented are misleading, in that they give an inflated impression of the real processing uncertainty in the EC method. Moreover, I think the total "processing uncertainty" in Fig. 12 is of no use even as a sensitivity metric, as it blends sensitivity to choices that are unclear with sensitivity to other choices that are very clear. (Similarly, the total uncertainty defined in Section 2.4.6 is of limited use because it blurs the distinction between stochastic half-hourly noise, which can be averaged out, and long-term systematic bias, which cannot.) So I would present instead (and show in Fig. 12) the flux sensitivities to the various individual processing choices. Then if the authors want to identify which processing choices are genuinely debatable and use their sensitivities to calculate a more meaningful overall processing uncertainty, they can do that. And then if they want to compare the magnitude of the systematic processing uncertainty (i.e. potential bias) to that of the random flux noise, they can do that too (But why? Over what noise averaging period is such a comparison meaningful?).

2. The distributions of lag times in Fig. 3b and 3d are concerning. Why the spike at 2.7 s, in the wing of a broad peak centered on 3.2 s? The spike seems to suggest

that the true lag was actually always 2.7 s, while the other retrieved lags were in error, perhaps due to some stochastic noise artifact. After all, if the true lag really were varying stochastically as suggested by the broad peak, then why would there be a preponderance of times when it was exactly 2.7 s? Was there perhaps a change in the experimental setup at some point during the measurement period? After seeing Figs. 3 and 4, I'm actually inclined to think that using constant lag is the most advisable option for these data. The authors instead recommend the DetLim method but do not justify that recommendation. In particular, it's unclear why the lag determined from $CO_2$ should ever be any worse than that determined from COS (except when the $CO_2$ flux crosses zero), given that both gases are measured by the same instrument and the $CO_2$ almost always has a higher signal to noise ratio.

3. Surprisingly, and despite the statement on lines 344 and 350, Fig. 7a seems to show that the COS cospectra don't seem to have any high-frequency signal loss, unlike the $CO_2$ cospectra. I can think no reason why that should be the case unless the large high-frequency instrument noise for COS is synchronizing by chance with high-frequency fluctuations in w. Given that the COS cospectra seem to match well with the temperature cospectra, it doesn't seem to make sense to use the $CO_2$ cospectral correction (based on the mismatch between the $CO_2$ and temperature cospectra) for COS. Unless perhaps Fig. 7a mistakenly shows COS cospectra after correction?

Technical Corrections

- line 2: "the recent development" should be "recent developments" - line 21: "not being" should be "is not" - line 22: "for radiation-dependency" should be "for the radiation-dependency" - lines 60-61: The word "respectively" doesn't make sense here, as there's nothing for the analyzers to be respective to. I recommend changing "at 10 Hz from Aerodyne Research (Billerica, MA, USA) and Los Gatos Research (San Jose, CA, USA), respectively" to "at 10 Hz, one from Aerodyne Research (Billerica, MA, USA) and one from Los Gatos Research (San Jose, CA, USA)". - line 64: I would delete "a basis of EC measurements" - line 112: "a home-made" should be just "home-made"

- line 149: "to some extent of weather changes" should be "to some extent weather changes" - line 156: "different" is superfluous, and so I would delete it - line 178: "others" should be "other reasons" - line 344: "compare Fig. 7a and 7b" should be "compare Fig. 7a and 7c" - lines 423 ff: "The uâĹŮ filtering is applied to conform the... does not make sense and I'm not sure exactly what you are trying to say here.

---

## Author Response (AR1)

Review response to discussion article "Towards standardized processing of eddy covariance flux measurements of carbonyl sulfide" by Kohonen et al.

Reviewer comments in black
Author response in purple
Altered text in the manuscript in italic
Changes made to the revised manuscript since the submission of the author response
Altered text in the revised manuscript (after author response to review comments)

**Reviewer #1: Georg Wohlfahrt**

General assessment: Kohonen et al. report on the effects of varying various post-processing steps required for eddy covariance COS flux measurements with the aim, as stated in the title, to standardize these. COS EC flux measurements are increasingly making their way into the literature as COS offers a novel means of constraining GPP and stomatal conductance. Yet, the necessary processing steps are way not as harmonized as is the case for CO2, potentially causing systematic bias between studies using different processing schemes, thus impeding synthesis activities. Overall I think this is a timely and relevant addition to the literature, which fits with the scope of the journal. I though also believe that the manuscript suffers from several issues, which will require significant changes, as detailed below.

Major comments:
(1) First, I have several formal issues with the manuscript: English style is often poor, which creates situations in which the intended meaning is not entirely clear (e.g. l. 32 the explanation of footprint limitations during stable stratification). Some of the formulations are too sloppy and thus misleading (e.g. l. 33 where "operation at high frequency" is mixed with "fast time response"). Some text is trivial or circular (e.g. l. 434-435), some of the concepts are wrong (e.g. l. 49) and some information is missing (e.g. legend of Fig. 11). Often some later, in-house studies are cited instead of the original papers. Finally, a mix of tenses is used when typically the past tense should be used to describe own results.
We will revise the manuscript and try to improve the english language, avoid circular text and repetition and only use the past tense. We cite the original papers as suggested. The caption of Fig. 10 (former Fig. 11) will be improved.
Manuscript has been revised and english language improved. Circular text and repetition has been reduced and only past tense is used in the text. All figure captions have been carefully checked to include all information.

(2) Novelty and justification of the study: In 2017 a methodological paper on COS EC flux measurement post-processing was published in the same journal (Gerdel et al.). The authors justify their paper mainly by stating that their analysis goes beyond this previous paper. While this is partially true (in particular the analyses on lag times is novel), I think the authors should follow what the title of the paper suggests and rather sell their work as contributing towards a standardization of COS EC flux post-processing routines. To this end, I suggest to synthesize, e.g. in a table, the various processing steps that were used by previously published studies as a starting point and use this as a backbone for their analysis and the resulting recommendations. This table would then summarize whether and if so how previous studies detrended their time series, how the lag time was found, how low/high-frequency response corrections were applied,

whether data were filtered for low u* (how were thresholds found) and which QC/QA was used. Following this suggestion requires at least the introductory section to be more or less completely re-written and would allow the paper to live up to what its title suggests and eventually become a reference for COS EC flux measurements.

Thank you for this suggestion. The Introduction section will be reorganized and partly rewritten. The study's objectives will be more clear. A table summarizing previous studies is a very good idea and will be implemented as Table 1 in the new version of the manuscript. We will revise the whole manuscript and extend the discussion to processing routines used in earlier studies.

The introduction section has been reorganized and partly rewritten, ending with a statement of study objectives. Table 1 summarizing processing steps used in previous studies has been added to the introduction.

(3) Vertical advection: This section is somewhat odd – the authors acknowledge that knowing the magnitude of vertical advection is meaningless unless the magnitude of horizontal advection is known as well, yet vertical advection is reported even though horizontal advection has not been quantified. Unless the authors can come up with a discussion of what their results on the sign and magnitude of vertical advection actually mean in the context of their study, I thus suggest removing the results on vertical advection and all text/material that pertains to it.

We have considered this point carefully and came to the conclusion to leave out the results regarding vertical advection. The reviewer has a good point and as we are aiming for harmonization of processing protocols - where vertical advection is not used - we decided to leave this section out of the revised manuscript.

Vertical advection was left out of the manuscript.

(4) Corrections for high-frequency flux loss: Comparing two different approaches is novel for COS, yet surprisingly none of the underlying results are shown – I suggest to expand this section.

We discuss and show the difference to the reference processing scheme and show in Table 2 the effect of spectral corrections to final fluxes. Histograms and PDFs of different spectral correction schemes will be added to the Supplement (Fig. S2) and daytime and night-time median fluxes added to Table 2 and discussed in the text. We will also add a figure on flux attenuation to Supplementary material Fig. S3 and scatter plot comparing the final fluxes to Fig. S7.

Table 2 summarizes the differences in the final fluxes with the different spectral correction methods (total median flux, daytime and night-time median fluxes). We added the two cospectral models to Fig. 4 of the revised version of the manuscript, and histograms (and PDF) of the final fluxes to Supplementary material Fig. S6, as well as a scatter plot comparing the final fluxes to Fig. S11.

(5) Changes of co-spectral peak frequency with stability: Among the results of this study is a figure comparing the changes in the co-spectral peak frequency with stability for the Horst model and this study. While interesting, this analysis and the results are not motivated in the introduction and are barely discussed. Again, unless the authors are able to come up with a discussion of what the observed differences mean for their study, I suggest removing this material (or possibly moving it into a supplement).

We will move the figure to supplementary material (Fig. S4), as suggested. Equations 15 and 16 of the former version will be moved to Methods-section and presented as equations 6 and 7 in the revised version of the manuscript.

The figure was moved to supplementary material Fig. S8 and all equations realted to high-frequency spectral corrections are reported in Sect. 2.4.3 of the revised manuscript.

(6) Gap-filling: This is an indeed novel aspect, however way underexploited by the authors. Only a single arbitrarily chosen gap-filling algorithm is tested, the authors and miss to put it to a true test and results of gap-filling (e.g. time course of estimated parameters and selected results illustrating gap-filling behaviour) are lacking.

A time series of gap-filling parameters will be shown in the supplementary material Fig. S5. We will add a diurnal plot of the measured flux compared to different gap-filling functions in Fig. 11 in the revised manuscript as well as residuals of different methods in Fig. S6 in the supplementary material. We are not going into further detail in comparing different gap-filling methods, as that would result in a whole new paper. Long-term budgets are not usually the interest of the COS community, as GPP calculations from COS often aim at understanding the $CO_2$ exchange dynamics rather than calculating long-term budgets. However, it is important to fill short gaps (individual 30 mins or a bit longer) to e.g. get diurnal variations. The presented gap-filling method is just one example that can be used for gap-filling COS data. We will add discussion in Section 3.6 (former Section 3.7): "*Three combinations of environmental variables (PAR, PAR and relative humidity, PAR and VPD) were tested using the gap-filling function Eq. 16. These environmental parameters were chosen because COS exchange has been found to depend on stomatal conductance, which in turn depends especially on radiation and humidity (Kooijmans et al., 2019). Development of the gap-filling parameters a, b, c and d over the measurement period is presented in the Supplementary material Fig. S5. While saturating function of PAR only captured the diurnal variation already relatively well, adding a linear dependency on VPD or RH made the diurnal pattern even closer to the measured one (Fig. 11). Therefore, the combination of saturating light response curve and linear VPD dependency was chosen. Furthermore, we chose a linear VPD dependency instead of a linear RH dependency due to smaller residuals in the former (Fig. S6).*"

Time series of gap-filling parameters and residuals of different gap-filling methods are shown in Figs. S10 and S9 in the supplementary material, respectively. Diurnal variation of the different gap-filling functions in July-August are shown in Fig. 7 of the revised manuscript. The revised text in Sect. 3.6 reads "*Three combinations of environmental variables (PAR, PAR and relative humidity, PAR and VPD) were tested using the gap-filling function Eq. 17. These environmental parameters were chosen because COS exchange has been found to depend on stomatal conductance, which in turn depends especially on radiation and humidity (Kooijmans et al., 2019). Development of the gap-filling parameters a, b, c and d over the measurement period is presented in the Supplementary material Fig. S10. While saturating function of PAR only captured the diurnal variation already relatively well, adding a linear dependency on VPD or RH made the diurnal pattern even closer to the measured one (Fig. 7). Therefore, the combination of saturating light response curve and linear VPD dependency was chosen. Furthermore, we chose a linear VPD dependency instead of a linear RH dependency due to smaller residuals in the former (Fig. S9).*"

(7) Li-6262 and QCL CO2 cross-covariance maximisation: An important detail of this study is the setup, which includes a closed-path IRGA that is used to measure CO2 and H2O concentrations (not clear whether from the same tube as the QCL). These data are used to account for the drift in the computer clocks acquiring the sonic (& IRGA) and QCL data, respectively. While this nicely shows the benefit of having a complementary suite of measurements at "super-sites" such as Hyytiälä, in my view the reliance on an additional instrument is a drawback of this study as it limits the applicability of the proposed approach at other sites where no additional IRGA or an open-path model or closed-path model with a short tube is deployed. Even more so, this approach is unnecessary, as there are simpler, software-based, solutions available to keep computer clocks in

a network synchronized. In addition, by aligning data this way, the authors ruin a truly independent means of cross-comparing the QCL CO2 and H2O fluxes. I thus think it would be useful to explore the possibility of aligning the data sets in time without the help of the IRGA data. This is possible by expanding the time window in which the lag determination algorithm searches for, as with computer clocks being reset only once a day, time shifts of several seconds may result (in both directions). The reliability of this approach may then be checked by comparing against lag times and fluxes calculated with the IRGA.

We thank the Referee for this point. However, we recognize that standard CO2/H2O flux towers measuring other gases than CO2 typically use QCLs, which are nowadays providing also H2O and CO2, beside the target gas (CH4, N2O, COS, etc). In our study we have taken advantage of this setup. We acknowledge that better synchronization approach should be used already in the data logging system and a short text will be added to Section 2.2. Moreover, we have considered the Referee's suggestion of using a larger time window for COS, but it would be problematic because the covariance peak is not always clear and with a large time window there might be multiple peaks (for example related to low frequency variations). Instead, we have tested the file combination maximizing the covariance of CO2 (QCL) and w, and this resulted in a very similar outcome as the previous method. The results are shown as histograms in the supplement (Fig. S1). We rewrote this part of the manuscript as: "*The following procedure was done to combine two data files of 30 min length (of which one includes sonic anemometer and LI-6262 data and the other includes Aerodyne QCLS data): 1) the cross-covariance of the two $CO_2$ signals (QCLS and LI-6262) was calculated 2) the QCLS data were shifted so that the cross-covariance of the CO2 signals was maximized. Note that this will result in having the same lag time for QCL and LI-6262. The time shift was a maximum of 10 seconds, with most varying between 0 s to 2 s during one day. It is also possible to shift the time series by maximizing the covariance of $CO_2$ and w, which will then already account for the lag time (Fig. S1) or combine files according to their time stamps and allow a longer window in which the lag time is searched. However, in this case it is important that the lag time (and time shift) is determined from $CO_2$ measurements only, as using COS data might result in several covariance peaks in longer time frames due to low signal-to-noise ratios and small fluxes.*"

The reported changes listed above were implemented in the revised manuscript.

(8) Conclusions: The authors should conclude with referencing against what processing steps have been used by previous studies and put their results into perspective with these, by highlighting critical steps and the need for further harmonization.

As suggested by the Referee, we will add more discussion related to previous studies in the "Results and Discussions" chapter. Moreover, in the "Conclusions" we will highlight more clearly the critical steps and needs for further harmonization.

More discussion in relation to previous studies was added to Sect. 3 Results and Discussion. We finish the Conclusion section with fnal remarks "*Our recommendation for time lag determination ($CO_2$ cross-covariance) differs from the most commonly used method so far (COS cross-covariance), while experimental high frequency spectral correction has been widely applied already before. Many earlier studies have neglected the storage change flux, but we emphasize its importance in the diurnal variation of COS exchange. In addition, we encourage implementing gap-filling to future COS flux calculations for eliminating short-term gaps in data.*"

Detailed comments:

l. 1-3: reformulate to better convey intended meaning "... growing in popularity with the aim of estimating gross primary productivity at ecosystem scale, however lack standardized protocols ..."
*Clarified. The sentences now read "Carbonyl sulfide (COS) flux measurements with the eddy covariance (EC) technique are becoming popular for estimating gross primary productivity. To compare COS flux measurements across sites, we need standardized protocols for data processing."*

l. 17: replace "due to the use" with something more suitable, e.g. "motivated by ..."
Corrected as suggested.

l. 20: ", but in contrast to CO2, COS is destroyed..."
Corrected as suggested.

l. 22: for readers not familiar with the LRU, talking about the radiation-dependency of the LRU without introducing the concept might be highly confusing
Text about LRU removed.

l. 29: "... the assumptions underlying the EC method ..."
Corrected as suggested.
This text has been removed in the revised manuscript.

l. 29-34 and l. 35-41: these two paragraphs are in my view too general to meaningfully add to the introduction
We reduced the paragraphs into one sentence: "*To meet the assumptions underlying the EC method, the site, setup design, and instrumentation need to be considered (Aubinet et al., 2000, 2012; Nemitz et al., 2018; Sabbatini et al., 2018).*"
This sentence has been removed from the revised manuscript.

l. 32: reformulate – what you likely mean is that the measurement height should be such that the footprint remains within the ecosystem of interest even during stable stratification
This sentence will be removed as we will harmonize the Introduction section and make it more compact.
The sentence has been removed.

l. 33: EC instruments need to have a fast time response, which is different from "operation at high frequency" (a slow-response sensor does not become suitable for EC only because its data are logged at 20 Hz), as it can be shown that fast-response measurements made every few seconds do not cause a systematic bias in the EC flux
Corrected.

l. 38: not sure this sentence applies universally to all closed-path analyzers and anyway I would think this is too much detail for the introduction – suggest to remove
Sentence removed.

l. 41: the first ones to report on this issue were Ibrom et al. (2007)

This sentence will be removed as we harmonize the Introduction section and make it more compact.
The sentence was removed from the revised manuscript.

l. 43-44: a rotation into the prevailing wind direction is only one step in the coordinate rotation; typically the aim is to align the coordinate system with the prevailing streamlines (2D or 3D) or with respect to some coordinate system that was established over a longer period (e.g. planar fit)
Corrected.

l. 49: the EC flux is fine – the problem is that it may represent a poor estimate of the surface-atmosphere exchange under these conditions
Corrected.

l. 53: in fact it was the following year (1999) that John Finnigan published a commentary on the Lee (1998) paper in which he demonstrated that correcting only for vertical advection is nonsense
All text and discussion regarding vertical advection will be removed from the revised manuscript.
All text and results regarding vertical advection were removed from the manuscript.

l. 56: if environmental data are lacking too, mean diurnal variation may be used as a last resort
Added mean diurnal variation.

l. 64: if the cross-covariance is flat, then a wrong lag time will not have a large effect
We agree that in case the covariance is flat, lag time does not make a large effect. But in the case when cross-covariance is not flat but noisy, and lag time determination is difficult, it affects the flux magnitude.

l. 66: I do not get the "However, . . ." which links to the previous sentence, which does not appear to make sense here
Removed "However,".

l. 69: Gerdel et al. did study lag determination (their section 3.1) and u*-filtering (their Fig. 6)
It was meant here that different methods for lag time determination have not been studied earlier. We have changed the sentence to "*Gerdel et al. (2017) describes the issues of different detrending methods, high-frequency spectral correction, lag time determination and u\* filtering. However, there has not been any study comparing different methods for lag time determination or high-frequency spectral correction in terms of their effects on COS fluxes.*"

l. 71: actually you do not discuss the "EC flux measurement setup" at all
Removed.

l. 71-73: the introduction should finish with a statement of objectives
We will revise the whole introduction section and end it with a clear statement of objectives: "*In this study, we compare different methods for detrending, lag time determination and high-frequency spectral correction. In addition, we compare two methods for storage change flux calculation, discuss the nighttime low turbulence problem in the context of COS EC measurements, introduce a method for gap-filling COS fluxes for the first time and discuss the most important sources of*

*random and systematic errors. Through the evaluation of these processing steps, we aim to settle on a set of recommended protocols for COS flux calculation.”*

l. 79: coordinate rotation
Corrected.

l. 78-88: does this have any relevance for this study?
We think it is important for the reader to know about the measurement site and its characteristics. Also, we want to mention that the same data has been published before in a paper focusing on different aspects than methodology.

l. 92: and sonic temperature
Corrected.

l. 95: flow rate through Li-6262, same pump as QCL, tube diameter, length, is the same tube as for the QCL?
The two instruments had their own inlet tubings. This is now clarified in the text and LI-6262 tubing information added: *“All measurements were recorded at 10 Hz frequency and were made with a flow rate of approximately 10 liters per minute (LPM) for the QCLS and 14 LPM for LI-6262, respectively. The PTFE sampling tubes were 32 m and 12 m long for QCLS and LI-6262, respectively, and both had an inner diameter of 4 mm. Two PTFE filters were used upstream of the QCLS inlet to prevent any contaminants entering the analyzer sample cell: one coarse filter (0.45 µm, Whatman), followed by a finer filter (0.2 µm, Pall corporation), at approximately 50 cm distance to the analyzer inlet. The Aerodyne QCLS used an electronic pressure control system to control the pressure fluctuations in the sampling cell. The QCLS was run at 20 Torr sampling cell pressure. An Edwards XDS35i scroll pump (Edwards, England, UK) was used to pump air through the sampling cell, while LI-6262 had flow control by a LI-670 flow control unit.”*

l. 106: is this the mean? what is the standard deviation?
The standard deviation was determined from the standard deviation of cylinder air measurements. Now clarified in the text: *“The standard deviation was 19 ppt for COS mixing ratios and 1.3 ppm for $CO_2$ at 10 Hz measurement frequency, as calculated from the cylinder measurements.”*
The sentence revised as : *“The standard deviation calculated from the cylinder measurements was 19 ppt for COS mixing ratios and 1.3 ppm for $CO_2$ at 10 Hz measurement frequency.”*

l. 111-112: given the precision of typical computer clocks, this will result in clock differences up to several seconds; important to add that most likely the Li-6262 data were acquired by and thus synchronized with the sonic anemometer through it's A/D input?!
Thanks for the comment. The logging system and data flow will be described in more detail in the revised manuscript.

l. 129-131: I am not sure I understand the first step – the QCL clock may be either delayed or advanced with respect to the clock of the sonic anemometer & IRGA and I thus do not understand why you shift the QCL time series by the lag time between w and IRGA CO2? In my understanding you could start off with the second step which actually aligns both time series.
The reviewer is correct, we have removed the first step. Sorry for the misunderstanding, now clarified in the text: *“The following procedure was done to combine two data files of 30 min length*

*(of which one includes sonic anemometer and LI-6262 data and the other includes Aerodyne QCLS data): 1) the cross-covariance of the two $CO_2$ signals (QCLS and LI-6262) was calculated 2) the QCLS data were shifted so that the cross-covariance of the $CO_2$ signals was maximized. Note that this will result in having the same lag time for QCL and LI-6262. The time shift was a maximum of 10 seconds, most often varying between 0 s to 2 s during one day. It is also possible to shift the time series by maximizing the covariance of $CO_2$ and w, which will then already account for the lag time (Fig. S1) or combine files according to their time stamps and allow a longer window in which the lag time is searched. However, in this case it is important that the lag time (and time shift) is determined from $CO_2$ measurements only, as COS data might result in several covariance peaks in longer time frames due to low signal-to-noise ratio and small fluxes."*

l. 131: might be worth showing this as a histogram in the supplement?
Time shift (computer drift + QCL lag) is now shown in the supplement Fig. S1.

l. 133-134: shouldn't this sentence come first in this section as this likely was the initial step? Or did the procedure described in l. 126-132 use data before despiking?
File combination was indeed done before despiking.

l. 142-143: not sure that computation time is a relevant issue nowadays with regard to coordinate rotation
Removed the note about computation times.

Fig. 1: the first rotation angle is typically the one that aligns the coordinate system along the main wind direction – why would that rotation be limited to less than 10 ∘ , which would mean rejecting fluxes from 340 ∘ ?
You're right, it was the second rotation angle, as mentioned in the text. Corrected in Fig. 1 now as well.

l. 200: in my memory, the first to propose this approach were Aubinet et al. (2000, 2001)
Already credited earlier, but added the reference here as well.

l. 201: a site-specific cospectral model was already used by Wohlfahrt et al. (2005)
In this line we only refer to Aubinet et al. (2000) as they introduced the technique and to De Ligne et al. (2010) as they compare different frequency response correction methods and make recommendations, but we will add Wohlfahrt et al. (2005) reference to earlier in the text.
The whole section on high-frequency spectral correction (Sect. 2.4.3) was revised. Text mentioned here was revised as "*The analytical correction was based on scalar co-spectra defined in Horst (1997) and the experimental approach was based on the assumption that temperature co-spectrum is measured without significant error and the normalized scalar co-spectra were compared to the normalized temperature co-spectrum (Aubinet et al., 2000; Wohlfahrt et al., 2005; Mammarella et al., 2009).*"

l. 213: which rotation angle?
Second rotation angle, now clarified in the text.

l. 232: but fluxes are available half-hourly – how do you come up with a half-hourly storage change estimate?

The storage change estimate from the concentration profile measurements are hourly and from EC measurements half-hourly. Concentration change used in the calculation is the change that occurred during that 30 min (cafter – cbefore), hence we get half-hourly storage estimates from the EC setup. The profile measurements were subsampled to 30 min to correct for the storage change in measured 30 min fluxes.

l. 237-238: 5 hours sounds like a really long time to reduce random noise – in fact I would expect a 5 hour moving average to even average out true storage; how exactly did you calculate the one-point storage term?

This is the same time window that was used in smoothing profile measurements (Kooijmans et al., 2017) and we chose the same time window for smoothing EC data for better comparison. If a shorter time window is used (e.g. 1 hour), the diurnal shape of the storage change flux does not change but noise of COS storage change flux increases almost three-fold (see figure below). One-point storage term was calculated by assuming that the concentration change in time is uniform from the measurement height to ground:

FCOSstor = h*ΔC/Δt

Where h is the measurement height (in m) and ΔC/Δt is the concentration change in time (in mol m-3 s-1) at the measurement height.

Below is an example of storage change flux when 1 hour moving average is used for smoothing concentration measurements. Median diurnal variation of COS storage change flux is not different from the 5 hour moving average, but variation has increased notably.

[Figure]

l. 264: clearly here only Lee (1998) is to credit with this approach

All text related to vertical advection will be removed from the revised manuscript.

All text related to vertical advection were removed from the revised manuscript.

l. 270: "missing CO2 fluxes"?

Corrected.

l. 282: add units
Units added.

l. 284-286: isn't this a repetition from above?
Yes, we will incorporate all the relevant information into the methods section.
All relevant information is given in the Methods section and repetition removed in the revised manuscript.

l. 288-289: all measurements are characterised by noise to a certain degree . . .
Yes, we do agree and we have rephrased it as "*As COS measurements are often characterized by low signal-to-noise ratio, the maximization of the absolute value of cross-covariance may determine the lag time from a local maxima, as demonstrated in Fig. 2.*"
This text, along with the former Fig.2, were removed from the revised manuscript.

l. 289-290: I do see several local minima in Fig. 2, but not that any of the tested algorithms gets stuck in one of these
We noticed that there was an artefact related to lag time calculation, as the lag time results were obtained from a lag time optimization tool. In the revised manuscript, we are using the lag times calculated based on the covariance maximization.
In the previous version of the manuscript, we used lag time optimization tool that replaces the lag time with a mean lag time if lag time is detected on the window border. We have reprocessed all fluxes without using the lag time optimization tool. Instead, we are now using a different method for determining the lag time:
"*The time lag between w and COS signals was determined using the following five methods:*
*1) From the maximum difference of the cross-covariance of the COS mixing ratio and w (*
$\overline{w'\chi_{cos}'}$ *) to a line between covariance values at the lag window limits (referred hereafter as*
*COS lag ). This applies also to other covariance methods explained below, and prevents the time lag to be exactly at the lag window limits.*"

l. 298-299: so what? How does that sentence relate to your results?
Thanks for the comment. The sentence will be rephrased accordingly.

l. 305: reformulate – I guess that what you mean is that with the DetLim method, the COS lag was selected in 40 % of all cases (while the CO2 lag was chosen in 60 %)
Reformulated as "*By using the DetLim lag method, the COS lag time was estimated for 40 % of cases from the* $\overline{w'\chi_{cos}'}$ *covariance maximization, while the $CO_2$ lag was used as proxy for the COS lag in about 60 % of cases.*"
After reprocessing all data, the phrase reads: "*By using the DetLim$_{lag}$ method, the COS time lag was estimated for 54 % of cases from COS$_{lag}$ , while the CO2$_{lag}$ was used as proxy for the COS time lag in about 46 % of cases.*"

l. 305: can you compare lag times of COS and CO2 both fluxes are clearly higher than the flux detection limit? Is there a systematic difference between the two (e.g. COS lag time always longer

than CO2) and if so could the DetLim Method be improved by adding this offset to the CO2 lag instead of just using the CO2 lag?

We tested this idea, but there was no systematic offset found between the lag times.

l. 309: why did you choose the cumulative flux as a metric? Wouldn't the cumulative flux potentially be affected by compensating effects (over- and underestimation during certain conditions resulting in similar cumulative fluxes)? Unlike for CO2, I also do not see much nee to calculate a daily or longer term budget for COS; I also do not see how you can get units of nmol/m2s for COS as for a cumulative flux you need to integrate over time, i.e. multiply the half-hourly flux by 1800 s and then sum these up this then yields molar units per m2 and time period over which the cumulative was calculated – same for CO2.

It is not a cumulative sum as used in budgets, thus not multiplied with 1800s and has units of nmol m-2s-1, i.e. just simply cumulative sum of all measured (NOT gap-filled!) fluxes. We have considered the reviewer's comment on compensating over- and underestimations in cumulative sum and chose to use median fluxes instead. Median fluxes are also affected by the compensating over- and underestimations but not as much to differences in missing data. We will add overall median fluxes as well as separate night-time and daytime median fluxes of all different processing schemes to Table 2.

All median fluxes (total, daytime and night-time, were implemented in Table 2 of the revised manuscript. All medians are calculated to periods when fluxes from all different processing schemes are available, to avoid systematic biases.

Table 1: instead of repeating the values for the reference three times, list it only once?!

Reference fluxes will be added to the table caption and removed from the table.

Rerefence fluxes were added to the table caption and removed from Table 2.

l. 315: how do you know this difference is large on an annual scale? What is the basis for this recommendation?

Changed the sentence to "*This difference might become important in the annual scale, and we recommend using the detection limit method in lag time determination of small fluxes, as in Nemitz et al. (2018)*" for clarity.

After reprossing all data and considering also the previous results, we landed on recommending the CO2 lag instead: "*This difference might become important in the annual scale, and as the most commonly used covariance maximization method does not produce a clear time lag distribution for DetLim$_{lag}$ or COS$_{lag}$, we recommend using the CO2$_{lag}$ for COS fluxes, as in most ecosystems the CO$_2$ cross-covariance with w is more clear than the cross-covariance of COS and w signals.*"

l. 335-336: "low-frequency corrections"

Corrected as suggested.

l. 336-337: combine both sentences

Combined sentences.

l. 337-338: this is kind of trivial

While this recommendation may be stating the obvious, it is not always carried out in field practice and therefore is worth emphasizing. We decided to keep the sentence, as it adds to the discussion.

l. 344: what is the "normal" CO2 cospectrum? Fig. 7a shows the COS cospectrum, 7b the COS power spectrum – this sentence does not make sense

Corrected the figure reference to 7a and 7c.

The correct figure reference is Fig. 4a and c in the revised manuscript.

Fig. 7: are the (co)spectra in any way filtered for stability or really averaged over the entire month? I suggest to remove the sub-grid lines in all four panels as otherwise these are too busy and become blurry

All the spectra are filtered for stability. Added a sentence to the (co)spectra figure's caption "*All data are filtered for stabilities -2 < z/L < -0.0625 and COS data only accepted when the covariance was higher than three times the random error due to instrument noise (Eq. 1).*" Gridlines will be removed.

Gridlines were removed. We added the model cospectra from the experimental and Horst (1997) methods to the figure and the caption now reads: "*Cospectrum and power spectrum for COS (a and b, respectively) and $CO_2$ (c and d) in July 2015. All data were filtered by the stability condition -2 < ζ < -0.0625 and COS data were only accepted when the covariance was higher than three times the random error due to instrument noise (Eq. 1). The cospectrum models by experimental method and Horst (1997), that were used in the high-frequency spectral correction, are shown in grey continuous and dashed lines, respectively.*"

l. 349: in Fig. 7 you are using normalized frequency, thus no units

Changed the text from "3 s" to "*normalized frequency higher than 3*"

l. 350: while the attenuation is clearly visible for CO2, it does not show well for COS, whose cospectrum mostly overlaps with the sensible heat cospectrum

Cospectrum is now plotted for only those times when COS flux surpasses the random noise. While there is still quite a lot of noise in the high frequency end of COS cospectrum, the attenuation is now more visible. Probably the instrument random noise was overlapping with sensible heat cospectrum by chance.

l. 350-351: how was this calculated?

Thanks for the comment. We will add more details in Section 2.4.3. In practice the response time $\tau_s$ was determined by fitting a sigmoidal function $1/(1+(2\pi f \tau_s)^2)$ to the ratio between ensemble averaged CO2 and T cospectra.

The modified text regarding response time calculation now reads:

"*In both approaches (analytical and experimental), the time constant $\tau_s$ was empirically estimated by fitting the transfer function $T_{ws}(f)$ to the normalized ratio of cospectral densities*

$$T_{ws} = \frac{N_\theta \, Co_{ws}(f)}{N_s \, Co_{w\theta}(f)} \qquad (10)$$

*where $N_\theta$ and $N_s$ are normalization factors and $Co_{ws}$ and $Co_{w\theta}$ the scalar and temperature cospectra, respectively. The estimated time constant was 0.68 s for the Aerodyne QCLS and 0.35 s for LI-6262.*"

l. 353: indeed this is as expected . . . remove?

Removed the sentence.

l. 355-357: I think it would be instructive to show at least one characteristic example comparing the experimental and analytical frequency response correction approaches as otherwise this remains a "black box" for the reader

The two correction methods are now fully compared by adding median daytime and night-time fluxes to Table 2, histogram of fluxes to Supplementary material Fig. S2 and stability categorized flux attenuation versus wind speed to Supplementary material Fig. S3. Otherwise, we do not understand what the referee means by saying "to show at least one characteristic example".

We have added the two different cospectral models to Fig. 4 of the revised manuscript. In addition, a histogram of fluxes is shown in Fig. S6 and flux attenuation factor versus wind speed in Fig. S7 in the supplementary material.

l. 358: while I understand that the experimental frequency response correction approach is part of the standard against which the comparison is made, I think (i) that the no frequency response correction scenario is useless as we know that this leads to a bias and (ii) instead it would be useful to compare the magnitude of the correction between the analytical and experimental approach in order to understand how much of a difference it makes whether one or the other correction is used

The "no correction" option demonstrates how much high frequency corrections affect the final fluxes and is thus left to the analysis. Magnitude of the correction is demonstrated in Table 2 as total median fluxes and daytime and night-time median fluxes. Histogram of the resulting COS fluxes is added to Supplementary material Fig. S2 and flux attenuation versus wind speed to Fig. S3. Final fluxes are also compared as a scatter plot in Fig. S7.

Histogram of the resulting COS fluxes is added to Supplementary material Fig. S6 and flux attenuation versus wind speed to Fig. S7. Final fluxes are also compared as a scatter plot in Fig. S11.

l. 364-369: this section comes a bit as a surprise as it never has been mentioned as a goal before to do this kind of comparison and also lacks a proper discussion – without discussion I rather suggest to remove Fig. 8 and the corresponding text; one point of discussion might be how much the difference in the cospectral reference models contributes to the differences between the analytical and experimental approach; btw., the results in Fig. 8 are qualitatively consistent with Fig. 11 of Wohlfahrt et al. (2005)

As this was not a very important part of our analysis, we have now moved the figure to supplementary material (Fig. S4), as suggested, and removed the corresponding text. Equations 15 and 16 of the former version were moved to Methods-section and are now presented as equations 6 and 7 in the present version of the manuscript.

Former Fig. 8 was moved to supplementary material Fig. S8 and all text to Methods section.

l. 371-377: here I have the impression that you not describing the actual magnitude of the correction, but rather what we might expect based on one example shown in Fig. 5?!

This correction is based on theoretical transfer functions by Rannik & Vesala (1999), and the range of magnitude of this correction is maximum 15 %. The correction is not related to setup or site specific, it is a general and theoretical correction, and expected to be similar to Rannik & Vesala (1999) study. It does not make sense to only correct for low frequency loss in this study to check the actual magnitude of the correction. We will remove this section from the revised manuscript and move Fig. 5 to supplementary material.

Discussion on low frequency spectral correction was left out of the revised manuscript.

l. 372-374: this expectation would only be justified if the algorithm used for correcting for flux loss at lower frequencies would "know" of the noise, which I expect it does not

Exactly, it does not know of the noise and that's why cannot make large corrections. We have decided to leave out the result section on low frequency spectral corrections.

Low frequency spectral correction was left out of the revised manuscript.

Figure 9: to what period do the data shown refer to?

To the whole measurement period. Clarified in the figure caption "*...during the measurement period 26 June to 2 November 2015.*"

l. 400-401: again, this is well established

We agree with the Reviewer, yet storage change flux is neglected in most of COS EC studies (Table 1). Thus, we feel it is important to emphasize that for diurnal variation it cannot often be neglected. We revised the sentence to emphasize this point: "*In conclusion, the storage change fluxes are not relevant for budget calculations – as expected – and have not been widely applied in previous COS studies (Table 1). However, storage change fluxes are important in the diurnal scale to account for the delayed capture of fluxes by the EC system under low turbulence conditions.*"

Sentence now reads: "*In conclusion, the storage change fluxes are not relevant for budget calculations – as expected – and have not been widely applied in previous COS studies (Table 1), even though storage change flux measurements are mandatory in places where the EC system is placed at a height of 4 m or above according to the ICOS protocol for EC flux measurements (Montagnani et al., 2018). In addition, storage change fluxes are important in the diurnal scale to account for the delayed capture of fluxes by the EC system under low turbulence conditions.*"

l. 418: I would not call data erroneous – they simply do not fulfil the assumptions underlying our simplified model of surface-atmosphere exchange

Modified the sentence to "*...reliable tool to filter out data that is not representative of the surface-atmosphere exchange under low turbulence (Aubinet et al., 2010).*"

l. 421-423: with this reasoning, wouldn't it make sense to get rid of the concentration dependency by using the deposition velocity, i.e. the flux normalized with the concentration, instead of the flux itself? Or perform the analysis with data stratified by COS concentration to minimize the issue?

Thank you for this suggestion! We made the u* plot using the deposition velocity (EC flux + storage change flux normalized with concentration gradient), but the u* dependency did not disappear. We added the following text to the manuscript: "*However, we did not see $u_*$ dependency disappearing even with a concentration gradient-normalized flux, so the $u_*$ filtering is applied here normally to overcome the EC measurement limitations under low turbulence conditions.*"

The sentence now reads: "*However, as we did not see $u_*$ dependency disappearing even with a concentration gradient-normalized flux, the $u_*$ filtering is applied here normally to overcome the EC measurement limitations under low turbulence conditions.*"

[Figure]

l. 428-437: knowing that the u*-filtering needs to be applied on the sum of the storage and EC flux, why do you still give the numbers for the EC flux without storage?

There are quite many COS studies that use *u** filtering even though they neglect (or don't mention) the storage change flux: Asaf et al. 2013, Billesbach et al. 2014, Maseyk et al. 2014, Commane et al. 2015, Yang et al. 2018 (and not sure how it was done in Spielmann et al. 2019, as it is not mentioned). So even though this is somewhat trivial for EC measurements, it has not been implemented widely in COS studies, and is worth mentioning here.

l. 434-435: circular argument - without applying the storage correction it is pretty clear that the storage flux is clear that the storage flux is ignored

Reformatted the sentence to "*If fluxes are not corrected for storage before deriving the u* threshold, there is a risk of flux overestimation due to double accounting. The flux data filtered for low turbulence would be gap-filled, thereby accounting for storage by the canopy, but then accounted for again when the storage is released and measured by the EC system during the flushing hours in the morning (Papale et al. 2006).*"

Figure 11: also shows the vertical advection without being mentioned in the figure legend?!

Vertical advection was mentioned in the figure legend, but was missing from the caption. We have decided to leave out the section regarding vertical advection, according to the reviewer's suggestions, and thus removed also from this figure.

l. 442-444: is this a useful comparison? The cumulative COS flux must be less negative if missing data (which are generally negative) are not gap-filled! In order to evaluate the skill of the gap-filling algorithm the authors need to create artificial (but realistic) gaps in their time series (see CO2 flux literature to that end) and then compare the gap-filled against the measured (during the artificially created gaps) fluxes!

This comparison is just demonstrating how much change there is when using gap-filled versus non gap-filled fluxes in a cumulative sum, and the magnitude of course depends on the number of gaps in the data. We will add a diurnal plot of the gap-filling algorithm and measured fluxes (Fig. 11), a plot of the residuals of different gap-filling algorithms (Fig. S6) and a time series of the gap-filling parameters (Fig. S5) to the revised manuscript and supplementary material.
Diurnal variation of the measured COS flux and three different gap-filling algorithms are shown in Fig. 7 of the revised manuscript. Residuals of different gap-filling algorithms and time series of the gap-filling parameters are shown in Figs. S9 and S10, respectively.

l. 474-475: why do you recommend the experimental high-frequency correction approach? Is it more accurate? If so this needs to be demonstrated! What about the performance of the experimental approach in situations with low/noisy sensible heat cospectra and what about the effect of the QCL on the ratio central to the experimental approach?
We did not use single 30 min T cospectra for the model cospectra. Only the averaged cospectra (when the cospectra was good) were used for creating the model (see Sabbatini et al., 2018 for reference). A site specific cospectral model is suggested to be used instead of analytical ones in several studies (Aubinet et al. 2000, De Ligne et al. 2010), and especially in the new eddy covariance data processing protocols (Sabbatini et al., 2018; Nemitz et al., 2018). We will provide both Horst (1997) and experimental cospectral models in the cospectrum figure, together with the mean cospectrum.
The changes mentioned above were implemented in the revised manuscript.

**Reviewer #2**

Erkkilä (Kohonen) et al present a detailed and valuable analysis of the impact of various eddy covariance data processing options on the calculated ecosystem uptake of carbonyl sulfide (COS). They attempt to quantify the flux uncertainty deriving from the data processing, and they make recommendation for some of the options. The methods are sound and the manuscript is fairly well written and easy to follow. I do have some concerns about the analysis and interpretation of the results. In addition to the "major comments" of referee Wohlfahrt, with which I agree, I believe the paper could be made stronger by addressing the issues below.

Specific Scientific Comments
1. I disagree with the idea that the "processing uncertainty" reported here is actually an uncertainty in the calculated fluxes. Instead, it is a metric of the sensitivity of the calculated fluxes to different processing choices. Some of those choices are clearly better than others, and it doesn't make sense to calculate the fluxes in a way that is known to be pretty good and in another way that is known to be pretty bad and then say that the difference between the two ways is the uncertainty in the flux. In particular, the following data processing choices are obviously bad: (a) the COS lag and RM lag methods, (b) the RF 30s detrending method, (c) omitting high-frequency correction, (d) omitting the storage flux, (e) determining a u* filter threshold before including the storage flux, (f) omitting gap filling (for cumulative sums). None of those methods should be included when assessing methodological uncertainty, as there is no uncertainty about the fact that those methods should not be used. Thus the "processing uncertainties" presented are misleading, in that they give an inflated impression of the real processing uncertainty in the EC method. Moreover, I think the total "processing uncertainty" in Fig. 12 is of no use even as a sensitivity metric, as it blends

sensitivity to choices that are unclear with sensitivity to other choices that are very clear. (Similarly, the total uncertainty defined in Section 2.4.6 is of limited use because it blurs the distinction between stochastic half-hourly noise, which can be averaged out, and long-term systematic bias, which cannot.) So I would present instead (and show in Fig. 12) the flux sensitivities to the various individual processing choices. Then if the authors want to identify which processing choices are genuinely debatable and use their sensitivities to calculate a more meaningful overall processing uncertainty, they can do that. And then if they want to compare the magnitude of the systematic processing uncertainty (i.e. potential bias) to that of the random flux noise, they can do that too (But why? Over what noise averaging period is such a comparison meaningful?).

The uncertainty estimates presented here are based on the well established eddy covariance data processing protocols, presented in Sabbatini et al., 2018. We agree that processing choices listed in the comments are not as good as others. In this method, we calculate the fluxes using block averaging with planar fitting, block averaging with 2D wind rotation, linear detrending with planar fitting and linear detrending with 2D wind rotation. These are all very widely used processing schemes and not thought as obviously bad. The bad choices listed in the comment are not used for estimating processing uncertainty. The estimate of processing uncertainty is based on calculating the lowest and highest possible fluxes that come out from different (reliable) processing schemes and thus tells how much variation there can be in fluxes due to processing choices. The method is explained in detail in Section 2.4.6. However, as random error is dominating the total uncertainty, even a loose definition of the processing uncertainty would not inflate the total uncertainty to a large degree.

2. The distributions of lag times in Fig. 3b and 3d are concerning. Why the spike at 2.7 s, in the wing of a broad peak centered on 3.2 s? The spike seems to suggest that the true lag was actually always 2.7 s, while the other retrieved lags were in error, perhaps due to some stochastic noise artifact. After all, if the true lag really were varying stochastically as suggested by the broad peak, then why would there be a preponderance of times when it was exactly 2.7 s? Was there perhaps a change in the experimental setup at some point during the measurement period? After seeing Figs. 3 and 4, I'm actually inclined to think that using constant lag is the most advisable option for these data. The authors instead recommend the DetLim method but do not justify that recommendation. In particular, it's unclear why the lag determined from CO2 should ever be any worse than that determined from COS (except when the CO2 flux crosses zero), given that both gases are measured by the same instrument and the CO2 almost always has a higher signal to noise ratio.

Thank you for pointing this out. We checked the lag time issue and the spike at 2.7 s and winged shape were due to an artefact caused by a lag time optimization tool used in the final flux calculation. In the revised manuscript we will show the lag times determined directly from the maximum covariance, which doesn't have this artefact. Using CO2 lag time is probably not any worse from DetLim lag, and we found a more clear lag time distribution for CO2 lag than DetLim lag. The DetLim lag method was established in the eddy covariance data processing protocol by Nemitz et al., 2018 for small fluxes, but to our knowledge the method is implemented here for the first time. We will add a more clear statement in the revised manuscript.

In the previous version of the manuscript, we used lag time optimization tool that replaces the lag time with a mean lag time if lag time is detected on the window border, and this caused the weird behaviour in the lag time distributions. We have now reprocessed all fluxes without using the lag time optimization tool. Instead, we are now using a different method for determining the lag time:
"*The time lag between w and COS signals was determined using the following five methods:*
*1) From the maximum difference of the cross-covariance of the COS mixing ratio and w (*
$\overline{w'\chi_{\cos}'}$ *) to a line between covariance values at the lag window limits (referred hereafter as*

*COS lag ). This applies also to other covariance methods explained below, and prevents the time lag to be exactly at the lag window limits."*

After analysing data from this method and comparing to the results of the previous processing method, we have decided to recommend the $CO2_{lag}$ instead of the $DetLim_{lag}$. The reviewer has a good point saying "In particular, it's unclear why the lag determined from CO2 should ever be any worse than that determined from COS (except when the CO2 flux crosses zero), given that both gases are measured by the same instrument and the CO2 almost always has a higher signal to noise ratio." especially, when the two molecules are chemically very similar. We have taken this into account in the revised manuscript and rewrote the recommendation on lag time as "*This difference might become important in the annual scale, and as the most commonly used covariance maximization method does not produce a clear time lag distribution for DetLim$_{lag}$ or COS$_{lag}$, we recommend using the CO2$_{lag}$ for COS fluxes, as in most ecosystems the $CO_2$ cross-covariance with w is more clear than the cross-covariance of COS and w signals.*"

3. Surprisingly, and despite the statement on lines 344 and 350, Fig. 7a seems to show that the COS cospectra don't seem to have any high-frequency signal loss, unlike the CO2 cospectra. I can think no reason why that should be the case unless the large high-frequency instrument noise for COS is synchronizing by chance with high-frequency fluctuations in w. Given that the COS cospectra seem to match well with the temperature cospectra, it doesn't seem to make sense to use the CO2 cospectral correction (based on the mismatch between the CO2 and temperature cospectra) for
COS. Unless perhaps Fig. 7a mistakenly shows COS cospectra after correction?

We think that it was indeed by chance that the instrument noise was synchronizing with high frequency fluctuations. We have now made the same plot only for COS fluxes that surpass the random noise (covariance at least three times higher than the random noise) and the flux attenuation is more evident and closer to that of CO2.

Technical Corrections
- line 2: "the recent development" should be "recent developments"
Corrected

- line 21: "not being" should be "is not"
Corrected

- line 22: "for radiation-dependency" should be "for the radiation-dependency"
Corrected
This sentence was removed from the revised manuscript.

- lines 60-61: The word "respectively" doesn't make sense here, as there's nothing for the analyzers to be respective to. I recommend changing "at 10 Hz from Aerodyne Research (Billerica, MA, USA) and Los Gatos Research (San Jose, CA, USA), respectively" to "at 10 Hz, one from Aerodyne Research (Billerica, MA, USA) and one from Los Gatos Research (San Jose, CA, USA)".
We will revise the introduction based on reviewer comments, and have left this part out of the text to harmonize and make the introduction more compact.
The sentence was left out of the revised manuscript.

- line 64: I would delete "a basis of EC measurements"
Deleted

- line 112: "a home-made" should be just "home-made"

Corrected

- line 149: "to some extent of weather changes" should be "to some extent weather changes"
Corrected as suggested

- line 156: "different" is superfluous, and so I would delete it
Deleted

- line 178: "others" should be "other reasons"
Corrected

- line 344: "compare Fig. 7a and 7b" should be "compare Fig. 7a and 7c"
Corrected
Corrected as Fig. 4a and c in the revised manuscript.

- lines 423 ff: "The uâĹŮ filtering is applied to conform the. . . does not make sense and I'm not sure exactly what you are trying to say here.

[revised manuscript text omitted]

---

## Author Response (AR2)

Review response to discussion article "Towards standardized processing of eddy covariance flux measurements of carbonyl sulfide" by Kohonen et al.

Editor/ reviewer comments in black
Author response in blue
*Altered text in the revised manuscript in italic*

**Associate Editor Decision: Publish subject to minor revisions (review by editor)** (27 Apr 2020)
by Christof Ammann
Comments to the Author:
Dear authors,
the two Referees have reviewed the revised version of your manuscript and found significant improvements. However, according to their assessment (see the individual referee reports in the manuscript overview) there are still some shortcomings that need further improvement. Please address them duely in your further revision.
In addition, please check carefully, whether all supplementary material is cited in the main manuscript. Uncited material should be deleted.

Best regards
Christof Ammann
AMT Associate Editor

We have addressed the reviewers' comments and made revisions accordingly. We have checked that all supplementary material is cited in the main manuscript.

Review by Georg Wohlfahrt:

General comments:
It is great to see that the authors have ironed out most of the issues that I addressed in my review of the first version of this manuscript. I think the second version is much better and actually lives up to what the title suggests and makes a significant contribution to the literature. There are however two more major issues, in addition to a few minor ones, that need to be fixed before I can recommend publication in AMT.

(1) Cumulative sums with and without gap-filling: This is something that I have commented on before – it is misleading in my view to report cumulative sums over some period if there are gaps in the data. This is why need gap-filling! I am trying to illustrate my point with an example that we have a better feeling for than COS flux, namely precipitation: Average annual precipitation at Hyttiälä is around 700 mm – if due to data gaps the sum would be only 350 mm – would you think it is useful to report that number? What do we learn from that number apart from that a continuous record is required for deriving annual sums as otherwise reported numbers are biased? What the authors could do though is to compare the mean or median of the non-gap-filled and gap-filled data – this would tell them something about what gap-filling does to the distribution of the data, e.g. if it is mostly nighttime data that are missing than I would expect the mean/median to increase (i.e. become less negative).

*We agree with the reviewer and have now reported the medians of non-gap-filled and gap-filled data instead of cumulative fluxes in the revised manuscript. The revised mauscript now reads:*
*"The average of the corrected and gap-filled COS fluxes during the whole measurement period was -12.3 pmol m$^{-2}$ s$^{-1}$, while without gap-filling the mean flux was 14 % more negative, -14.3 pmol m$^{-2}$ s$^{-1}$. This indicates that most gap-filling was done for the night-time data, when COS fluxes are less negative than during daytime.*

*For $CO_2$, 41 % daytime $CO_2$ fluxes were discarded, while 67 % of fluxes were discarded during nighttime. Altogether comprising 53 % of all $CO_2$ flux data. $CO_2$ fluxes were gap-filled according to standard gap-filling procedures presented in (Reichstein et al., 2005). The average NEE after all corrections and gap-filling was -2.14 μmol m$^{-2}$ s$^{-1}$, while without gap-filling the mean flux was 39 % more negative, -3.53 μmol m$^{-2}$ s$^{-1}$. Similar to COS, $CO_2$ fluxes are also mostly gap-filled during night-time. As night-time $CO_2$ fluxes are positive, gap-filled fluxes include more positive values than non-gap-filled fluxes, thus making the mean flux less negative."*

(2) Gap-filling: This section has been considerably expanded, yielding interesting insights, but also a few concerns. First, I think the current presentation of these results is way too qualitative – we need hard facts on how the various gap-filling algorithms compare, i.e. the usual statistics and when comparing algorithms with differing numbers of parameters a metric that takes this into account needs to be used, e.g. the AIC. Along these lines, Fig. 7 would also profit from showing the uncertainty/range of the measured/simulated values. Second, Fig. 7 reveals a substantial diurnal bias, especially during the night and the first half of the day when the three algorithms grossly overestimate COS uptake (at 05:00 it looks like measured uptake is -11 pmol/m2s, while all three algorithms calculate an uptake of around -18 pmol/m2s), which interestingly does not show up in Fig. S9a which reports the largest bias at the highest radiation values.
Given the form of the algorithm:
Fcos = a*I/(I+b) + c*D + d
it would appear to me that a much better fit could be obtained by choosing a less negative d, which causes a parallel upward shift of the simulated values, and by decreasing the value of b in exchange in order to increase the response to radiation. Overall, the overestimation is less prominent during the afternoon, in particular when RH/VPD is included, which suggests this to be a behavior resulting from asymmetry in stomatal behavior, discussed already in Kooijmans et al. (2019) – the

addition of RH/VPD however appears unable to fully account for this. I think the authors should try to improve the diurnal bias and at least discuss it.

We thank the reviewer for this comment. We have now optimized the gap-filling parameters so that parameter *d* was set to be the median night-time flux over the 15-day period in which the parameters were estimated, and other parameters were optimized after that. This shifted especially the night-time and early morning gap-fill function closer to measured COS flux, although still not beign perfect. As a metric for deciding the best gap-filling function, we have decided to use both the residuals and RMSE of the different functions, following previous EC studies. The mean residuals and RMSE values are now reported in the manuscript. We have also included the 25[th] and 75[th] percentiles of the gap-filling functions and measurements in Fig. 7 as uncertainty range.

We added the following paragraphs to the revised manuscript:
*"Parameter d was set as the median night-time COS flux over the 15-day window and other parameters estimated accordingly."*

*"The mean residual of the chosen model was -0.54 pmol $m^{-2}$ $s^{-1}$ and root-mean-square error (RMSE) 18.7 pmol $m^{-2}$ $s^{-1}$, while saturating PAR function with linear RH dependency had a mean residual of -0.84 pmol $m^{-2}$ $s^{-1}$ and RMSE 19.3 pmol $m^{-2}$ $s^{-1}$, and saturating PAR function had a residual of 0.97 pmol $m^{-2}$ $s^{-1}$ and RMSE 22.8 pmol $m^{-2}$ $s^{-1}$."*

Detailed comments:
l. 13: what does „normally" mean?
Replaced by *"as usual in EC flux processing"*.
l. 14: isn't this a repetition from above?
Repetition removed from earlier in the text.
Table 1: what does the "-" mean – that the information is not provided? Please note that Spielmann et al. (2019) used the approach as outlined in Gerdel et al. (2017) and thus corrected for storage by assuming the concentration changes at the EC height to representative of all heights below the EC sensors.
Exactly, the "-" means that information was not provided in the research article. We have now clarified it in the caption:
*"Processing steps not specified in the original articles are marked with a hyphen."*
l. 90: "acquired by" instead of "gathered to"
Corrected as suggested.
l. 127: processing of what?
Clarified as *"flux data processing"*
l. 227: it is not so much of a "drop", but rather a "decrease"
Corrected as suggested.
l. 293: "accounts for" instead of "smooths"
Corrected as suggested.
l. 352: here and on many other accounts – be careful when using low/high or increase/decrease with negative fluxes as often you get the direction wrong, as in this line; maybe resort to saying things like less/more negative and alike
Thank you for pointing this out! We have corrected this and similar occasions to "less/more negative" or "flux magnitude increased"
l. 412: what does "normally" mean?
Corrected as:
*"However, as we did not see $u_*$ dependency disappearing even with a concentration gradient-normalized flux, the $u_*$ filtering is applied here as usual (Papale et al., 2006) to overcome the EC measurement limitations under low turbulence conditions."*
l. 467-468: I accept that the authors want to show the magnitude of the frequency response corrections and thus report fluxes without these, BUT they should not mix up the difference

between corrected and uncorrected values with the difference between the two spectral correction approaches, which they do in this sentence! In the first part you discuss differences between different lag time approaches, while in the second part the difference between uncorrected and corrected fluxes. I think you should report the difference between the two spectral corrections in the second part of the sentence, which is just 2.7 % and thus relatively minor.

We thank the reviewer for pointing out this misleading sentence. The sentence now reads:
*"Different time lag methods made a difference of maximum 15.9 % in the median COS flux while correcting for high-frequency spectral losses influenced the median flux by 14.2 %. Different methods used in high-frequency spectral correction resulted in 2.7 % difference in the median fluxes."*

Reviewer #2

Kohonen et al have made substantial changes to their manuscript to address the reviewer concerns, and I think the paper is much improved. It will make a useful addition to the literature. I have only a few remaining suggestions, noted below.

1. The bolding of recommended options in Fig. 1, while a good idea, seems a bit subtle to me. I would like to see also an additional row in Table 1, giving the options recommended by this paper, which would serve as a quick and easy summary.

Thank you for this suggestion! We have increased the font size of the recommended options in Fig. 1 to make them more visible. We have also added a row in Table 1 summarizing recommendations given in this study.

2. Regarding the flux uncertainties, I thank the authors for clarifying that only a subset of the considered methods were used in the assessment of processing uncertainty. I see now that the manuscript states that briefly in Section 2.4.6, but I think Section 3.7 needs to make it clear as well. Currently, after reading through several subsections about how this or that processing choice impacts the flux, the reader comes to the final Results subsection (3.7) and finds an overall comparison of the relative impacts of processing and random uncertainty on the flux. At this point the reader should be informed that many of the processing choices they have just been learning about are not included, and why.

This is a good point. In section 3.7 we added clarification about the processing schemes used in estimating the processing uncertainty:

*"The uncertainty due to different processing schemes (BA with 2D coordinate rotation, BA with planar fitting, LD with 2D coordinate rotation and LD with planar fitting, as described in Sec. 2.4.6) in the flux processing contributed 33 % to the total uncertainty of the COS flux, the rest was composed of the random flux uncertainty (Fig. 8)."*

3. The authors seem to have overlooked my related concern, that the "total uncertainty" defined in Section 2.4.6 (and discussed in Section 3.7 and displayed in Fig. 8) blurs the distinction between stochastic noise, which can be averaged out, and systematic bias, which cannot. As EC flux measurements are always aggregated in some way (e.g. via daily averages, annual sums, regressions, or models), the sum of the random and systematic uncertainties for a single EC flux measurement is not of any practical relevance. If a reader wants to quantify how systematic processing uncertainty will contribute to the overall uncertainty in a result derived from EC data, they must first calculate how the random uncertainty in each individual flux measurement propagates through their method of aggregation (i.e. how it averages down), and then compare the random uncertainty in the aggregate result to the systematic processing uncertainty. The values of random and processing uncertainty presented in the manuscript are useful for that, but their sum and their relative contributions to that sum are not. So the "total flux uncertainty" in the manuscript is at best irrelevant and at worst misleading as to the relative importance of the random and systematic uncertainties to real EC analyses. I would remove it from the paper, and I would also add text calling the reader's attention to the fact that the random uncertainty reported is for a single flux measurement, and that it will average down in real EC analyses while the systematic processing uncertainty will not.

The reviewer makes a really good point and we have addressed the issue in the revised manuscript. We decided to keep the 30 min flux uncertainties in the manuscript but have now emphasized that it holds only for 30 min flux values and not for time averaged fluxes, as random uncertainty decreases with time-averaging and processing uncertainty does not. We added uncertainties of time averaged fluxes into Section 2.4.6 and two new panels in Fig. 8 demonstrating the uncertainties when fluxes are averaged over different time periods. We have also added discussion about time-averaged flux uncertainty in Section 3.7.

In the methods section we added these texts about time averaged flux uncertainty:

*"These processing schemes are equally reliable but cause variability in fluxes."*

*"It should be noted, though, that this uncertainty estimate holds for single 30 min fluxes only. When working with fluxes averaged over time, the total uncertainty cannot be directly propagated to the long-term averages because the two uncertainty sources behave differently. The random uncertainty is expected to decrease with increasing number of observations while processing related uncertainty would not be much affected by time averaging. The random uncertainty of a flux averaged over multiple observations is obtained as*

$$\langle \epsilon_{rand} \rangle = \sqrt{\frac{\Sigma_{i=1}^{N}(\epsilon_{rand,i})^2}{N^2}}$$

*where N is the number of observations and $\epsilon_{rand,i}$ the random uncertainty of each observation (Rannik et al., 2016). Unlike the random uncertainty, the systematic processing uncertainty does not decrease when averaging fluxes over time. 
[revised manuscript text omitted]
} \dfrac{1.05 n/n_m}{(1 + 1.33 n/n_m)^{7/4}} & \text{for } \zeta \leq 0, \, n < 1 \\[2ex] \dfrac{0.387 n/n_m}{(1 + 0.38 n/n_m)^{7/3}} & \text{for } \zeta \leq 0 \, n \geq 1 \\[2ex] \dfrac{0.637 n/n_m}{(1 + 0.91 n/n_m)^{2.1}} & \text{for } \zeta > 0 \end{cases} \tag{6}$$

In the experimental approach, we solved the Eq. 2 numerically and used the fitting of the measured temperature cospectra to define a site-specific scalar cospectral model (De Ligne et al., 2010) as

$$\frac{fCo_{w\theta}(n)}{\overline{w'\theta'}} = \begin{cases} \dfrac{10.36 n/n_m}{(1 + 4.82 n/n_m)^{3.05}} & \text{for } \zeta \leq 0, \, n < 1 \\[2ex] \dfrac{1.85 n/n_m}{(1 + 3.80 n/
[revised manuscript text omitted]

---

## Author Response (AR3)

Editor comments in black
Author response in blue
*Text in the revised manuscript in itali*c

Comments to the Author:
In the revised version, the authors have addressed most of the referee comments in an adequate way. However, there are a few minor issues that need to be corrected before the manuscript is acceptable for publication. They are listed in the following.

Best regards
Christof Ammann
* * *
Line 4:
This is syntactically incorrect. Change to "... and we provide a method …"
Corrected as suggested.

Fig. 1: For a better readability of the paper, the frequently used abbreviations for the different processing options (e.g. BA, RF, etc.) should also be indicated in Figure 1.
We have added the commonly used abbreviations in Figure 1, as suggested.

Line 425-428:
This is not an adequate discussion of the problem in my view. Filtering itself does not lead to a bias but rather the subsequent gap filling. So it needs to be discussed whether the gap filling of calm nights (below the u* threshold) based on data of windy nights (above the u* threshold) leads to a bias in the gap-filled COS fluxes. If a dependency on the concentration gradient is not excluded, it would need to be taken into account in the gap filling procedure.
It is indeed the gap-filling after filtering and not the filtering itself, that is causing the possible bias. As we did not see the u* dependency to disappear with concentration gradient-normalized flux, we did not see it possible to take this potential low-turbulence effect into account in gap-filling. We clarified this point in the revised manuscript:
*"Thus, the assumption that fluxes do not go down under low turbulence conditions, as is the case for respiration of $CO_2$, does not necessarily apply to COS uptake. Gap-filling the $u_*$ filtered COS fluxes may therefore create a bias due to false assumptions, if the gap-filling is only based on data from periods with high turbulence. However, as we did not see $u_*$ dependency disappearing even with a concentration gradient-normalized flux, the $u_*$ filtering and proceeding gap-filling were applied here as usual (Papale et al., 2006) to overcome the EC measurement limitations under low turbulence conditions."*

Line 496-499 (see last comment of referee G. Wohlfahrt):
This corrected formulation is still misleading as it mixes (compares) the variation of different alternative lag methods with the systematic effect of the necessary high frequency correction. The 2.7% variation in high-frequency correction method is much smaller than the 15.9% variation in the lag methods. Therefore, the sentence "The largest differences in the final fluxes came from time lag determination and spectral corrections" is inadequate or misleading, since a flux without high-frequency correction cannot be considered as final.
The editor makes a good point, and the sentence was indeed misleading. The section was reorganized as: *"The largest differences in the final fluxes came from time lag determination and detrending. Different time lag methods made a difference of maximum 15.9 % in the median COS flux. Different detrending methods, on the other hand, made a maximum of 6.2 % difference in the median COS flux while it was more important for $CO_2$ (10.7 % difference between linear detrending and block averaging). Omitting high-frequency spectral correction resulted in a 14.2 % lower*

*median flux, while different methods used in high-frequency spectral correction resulted in only 2.7 % difference in the final median fluxes. We suggest to use $CO_2$ time lag for COS flux calculation so that potential biases due to low signal-to-noise ratio of COS mixing ratio measurements can be eliminated. $CO_2$ mixing ratio is measured simultaneously with COS mixing ratio with the Aerodyne QCLS and in most cases has a higher signal-to-noise ratio and more clear cross-covariance with w than COS. Experimental high frequency correction is recommended for accurately correcting for site specific spectral losses. We recommend comparing the effect of different detrending methods on the final flux for each site separately, to determine the site and instrument specific trends in the raw data."*

Reviewer#2, Comment 2 (Line 469):
The author response to this comment was not fully satisfying. The authors should explain why they only used the coordinate rotation and the detrending effect to quantify the processing uncertainty.

The idea of the processing uncertainty is to calculate the fluxes with all possible processing schemes, but as that would lead to hunderds of processing step combinations, we have settled to the ones that give the most variation in the fluxes, but are equally realiable methods. Originally this suggestion comes from the flux processing protocol by Sababtini et al., (2018), where they give processing protocols for ICOS flux measurements and suggest to use coordinate rotation and detrending as the steps that give the largest effect on fluxes. We have now added also high-frequency spectral correction as one of the steps considered here and added the Horst (1997) method in addition to the experimental correction method (now leading to altogether 8 processing combinations). We have used the coordinate rotation, detrending and spectral correction method in the processing uncertainty estimation because these methods (BA, LD, planar fitting, 2D rotation, experimental correction and Horst (1997) correction) are equally reliable (unlike e.g. using all different lag time methods) but cause the most variation in the final flux. This is also demostrated in Table 2, where we have now added the effect of 2D coordinate rotation as one column.

In methods Section 2.4.6, we revised the manuscript as:
*"This uncertainty was estimated as*
$$\epsilon_{proc} = (max(F_{c,j}) - min(F_{c,j}))/\sqrt{12} \qquad\qquad (13)$$
*where $F_{c,j}$ is the flux calculated according to j=1,...,N different processing schemes and N is the number of possible processing scheme combinations that are equally reliable but cause variability in fluxes. For simplicity, the processing steps we considered here are detrending, coordinate rotation and high-frequency spectral correction, leading to N = 8 processing schemes: BA with 2D coordinate rotation and experimental spectral correction, BA with 2D rotation and Horst (1997) spectral correction, BA with planar fitting and experimental correction, BA with planar fitting and Horst (1997) correction, LD with 2D rotation and experimental correction, LD with 2D rotation and Horst (1997) correction, LD with planar fitting and experimental correction and LD with planar fitting and Horst (1997) correction."*

In results Section 3.7 we wrote:
*"The uncertainty due to chosen processing scheme was determined from a combination of eight different processing schemes, as described in Sec. 2.4.6, that were equally reliable but caused most variation in the final COS flux (Table 2). This processing uncertainty contributed 36 % to the total uncertainty of the 30 min COS flux, while the rest was composed of the random flux uncertainty (Fig. 8). For the $CO_2$ flux uncertainty, the processing was more important than for COS (48 %), but the random uncertainty was still dominating the combined flux uncertainty."*

Note that Fig. 8 and some numbers in the text were updated due to this change in the uncertainty estimation.

Line 510:

Change to: "When averaging fluxes over time, the relative random uncertainty …"

Corrected as suggested.

Line 511:

[revised manuscript text omitted]